**Future inhibition of ecosystem productivity by increasing wildfire pollution**
**over boreal North America**
Xu Yue[1], Susanna Strada[2], Nadine Unger[3], Aihui Wang [4]
[1] Climate Change Research Center, Institute of Atmospheric Physics, Chinese Academy of
Sciences, Beijing 100029, China
[2] Laboratoire des Sciences du Climat et de l'Environnement, L'Orme des Merisiers - Bat 712,
91191 Gif Sur Yvette, France
[3] College of Engineering, Mathematics and Physical Sciences, University of Exeter, Exeter,
EX4 4QE, UK
[4] Nansen-Zhu International Research Centre, Institute of Atmospheric Physics, Chinese
Academy of Sciences, Beijing 100029, China
*Corresponding author*:
Xu Yue
Telephone: 86-10-82995369
Email: xuyueseas@gmail.com
*Keywords*: wildfire emissions, ozone, aerosols, net primary productivity, climate change,
diffuse fertilization effect, carbon loss, earth system modeling

28                                              **Abstract**

Biomass burning is an important source of tropospheric ozone ($O_3$) and aerosols. These air
pollutants can affect vegetation photosynthesis through stomatal uptake (for $O_3$) and light
scattering and absorption (for aerosols). Wildfire area burned is projected to increase
significantly in boreal North America by the midcentury, while little is known about the
impacts of enhanced emissions on the terrestrial carbon budget. Here, combining site-level
and satellite observations and a carbon-chemistry-climate model, we estimate the impacts of
fire emitted $O_3$ and aerosols on net primary productivity (NPP) over boreal North America.
Fire emissions are calculated based on an ensemble projection from 13 climate models. In the
present day, wildfire enhances surface $O_3$ by 2 ppbv (7%) and aerosol optical depth (AOD) at
550 nm by 0.03 (26%) in the summer. By midcentury, area burned is predicted to increase by
66% in boreal North America, contributing more $O_3$ (13%) and aerosols (37%). Fire $O_3$
causes negligible impacts on NPP because ambient $O_3$ concentration (with fire contributions)
is below the damage threshold of 40 ppbv for 90% summer days. Fire aerosols reduce surface
solar radiation but enhance atmospheric absorption, resulting in enhanced air stability and
intensified regional drought. The domain of this drying is confined to the North in the present
day, but extends southward by 2050 due to increased fire emissions. Consequently, wildfire
aerosols enhance NPP by 72 Tg C $yr^{-1}$ in the present day but decrease NPP by 118 Tg C $yr^{-1}$
in the future, mainly because of the soil moisture perturbations. Our results suggest that
future wildfire may accelerate boreal carbon loss, not only through direct emissions
increasing from 68 Tg C $yr^{-1}$ at present day to 130 Tg C $yr^{-1}$ by midcentury, but also through
the biophysical impacts of fire aerosols.


## 1 Introduction

Wildfire area burned is increasing in recent decades over North America boreal regions (Stocks et al., 2002; Kasischke and Turetsky, 2006). Fire activity is closely related to weather conditions and large-scale atmospheric oscillations (Gillett et al., 2004; Duffy et al., 2005), and is projected to increase significantly in the future due to climatic changes (Flannigan et al., 2005; Balshi et al., 2009; Groot et al., 2013; Wang et al., 2015). More area burned and the consequent fire emissions are accelerating carbon loss in boreal North America (Bond-Lamberty et al., 2007; Turetsky et al., 2011). Meanwhile, fire-induced air pollution, including ozone ($O_3$) and aerosols, is predicted to increase in boreal and downwind regions by midcentury (Yue et al., 2013; Yue et al., 2015). Wildfire emissions have large impacts on air quality (Wotawa and Trainer, 2000; Morris et al., 2006), weather/climate conditions (Randerson et al., 2006; Zhao et al., 2014), and public health (Zu et al., 2016; Liu et al., 2017). However, little is known about how these pollutants affect ecosystem carbon assimilation, and how this impact will change with the increased wildfire activity in the future.

Surface $O_3$ causes damages to photosynthesis through stomatal uptake (Sitch et al., 2007). In the present climate state, fire-induced $O_3$ enhancements are predicted to reduce net primary productivity (NPP) in the Amazon forest by 230 Tg C yr$^{-1}$ (1 Tg = $10^{12}$ g), a magnitude comparable to the direct release of $CO_2$ from fires in South America (Pacifico et al., 2015). The aerosol effects are more uncertain because both positive and negative feedbacks occur. Appearance of aerosols increases diffuse light, which is beneficial for shaded leaves in the lower canopy. Consequently, photosynthesis of the whole ecosystem will increase as long as the total light availability is not compromised (Kanniah et al., 2012). Rap et al. (2015) estimated that biomass burning aerosols increase Amazon NPP by 78–156 Tg C yr$^{-1}$, which offsets about half of the damage caused by fire $O_3$ (Pacifico et al., 2015). In contrast, strong light attenuation associated with high aerosol loading may decrease canopy photosynthesis (Cohan et al., 2002; Oliveira et al., 2007; Cirino et al., 2014). Furthermore, the aerosol radiative effects indirectly influence ecosystem productivity through concomitant meteorological perturbations that are only beginning to be examined (Yue et al., 2017).

Future wildfire activity is projected to increase over boreal North America but with large
uncertainties (Flannigan et al., 2005; Tymstra et al., 2007; Girardin and Mudelsee, 2008;
Nitschke and Innes, 2008; Amiro et al., 2009; Balshi et al., 2009; Bergeron et al., 2010;
Wotton et al., 2010; de Groot et al., 2013; Wang et al., 2016). For example, Amiro et al.
(2009) predicted an increase of 34% in Canadian area burned for a $2 \times CO_2$ scenario (2040-
2060) relative to a $1 \times CO_2$ condition (1975-1995), using the Canadian Fire Weather Index
(CFWI) and output from Canadian global climate model (CGCM) version 1. Balshi et al.
(2009) projected that area burned in boreal North America would double by the year 2045-
2050 relative to 1991-2000, using the Multivariate Adaptive Regression Splines (MARS)
approach and meteorological output from CGCM version 2. The increasing rate in Balshi et
al. (2009) is higher than that in Amiro et al. (2009), indicating substantial uncertainties in fire
projections originating from both fire models and simulated future climate. However, even
with the same fire models and climate change scenario, large uncertainties (in both
magnitude and signs) are found in the projection of area burned among individual climate
models (Moritz et al., 2012; Yue et al., 2013). The multi-model ensemble approach has
shown superior predictability over single models in historical climate simulations (Flato et al.,
2013) and near-term climate predictions (Kirtman et al., 2014), and has been used as a
standard technique to assess changes of climate variables in the long-term projections
(Collins et al., 2013). Following this strategy, Yue et al. (2015) used output from 13 climate
models to drive fire regression models and predicted an average increase of 66% in boreal
area burned at 2046-2065 relative to 1981-2000 under the IPCC A1B scenario (Solomon et
al., 2007). Yue et al. (2015) further calculated that the wildfire emission increase by the
2050s would increase mean summertime surface $O_3$ by 5 ppbv in Alaska and 3 ppbv in
Canada. The study found regional maximum $O_3$ enhancements as high as 15 ppbv, suggesting
the potential for possible vegetation damage and land carbon loss due to the enhanced boreal
fire-related air pollution. Wildfire aerosols are also expected to increase significantly but not
predicted in Yue et al. (2015).

In this study, we quantify the impacts of $O_3$ and aerosols emitted from boreal wildfires on the
land carbon uptake in North America in the present climate state and in the future world at
2050, taking advantage of the ensemble projection of future wildfire emissions by Yue et al.
(2015). The major chain we investigate includes i) generation of aerosols and surface ozone
from wildfire emissions and ii) impact of fire-emitted aerosols and ozone on plant
photosynthesis through physical and biogeochemical processes (Fig. 1). We first analyze
relationships between gross primary production (GPP) and aerosol optical depth (AOD) at
550 nm over the boreal regions based on observations. We then perform a suite of Earth
system model simulations using NASA GISS ModelE2 that embeds the Yale Interactive
Terrestrial Biosphere model (YIBs), a framework known as ModelE2-YIBs (Yue and Unger,
2015). Future projections of wildfire emissions from Yue et al. (2015) are applied as input to
ModelE2-YIBs model to project fire-induced $O_3$ and aerosol concentrations in the 2010s and
2050s. The impacts of the boreal fire $O_3$ on forest photosynthesis are predicted using the flux-
based damage algorithm proposed by Sitch et al. (2007), which has been fully evaluated
against available $O_3$ damage sensitivity measurements globally and over North America (Yue
and Unger, 2014; Yue et al., 2016; Yue et al., 2017). Fire aerosols induce perturbations to
radiation, meteorology, and hydrology, leading to multiple influences on the land carbon
uptake. Sensitivity experiments are performed using the YIBs model in offline mode to
isolate the contributions of changes in the individual meteorological drivers.


**2  Materials and methods**

**2.1 Observed GPP-AOD relationships**

Following the approach by Strada et al. (2015), we investigate the GPP sensitivity to diffuse
radiation and AOD variability in boreal regions. First, we identify study sites in Canada and
Alaska from the AmeriFlux (AMF) network (http://ameriflux.lbl.gov/). There are much fewer
boreal sites than those in temperate regions. We select AMF sites providing hourly (or half-
hourly) simultaneous measurements of GPP (non gap-filled) and photosynthetically active
radiation (PAR, total and diffuse) for at least 3 consecutive years. Only two Canadian sites
meet the criteria: Groundhog River (CA-Gro, 82.2°W, 48.2°N), a mixed forest (MF), and
Quebec Mature Boreal Forest Site (CA-Qfo, 73.4°W, 49.7°N), an evergreen needleleaf forest
(ENF). At the two selected sites, we calculate the Pearson's correlation coefficients between
half-hourly GPP and different components of PAR. In total, we select 2432 and 3201 pairs of
GPP and PAR measurements at CA-Gro and CA-Qfo, respectively. We then apply
instantaneous Level 2 Collection 6 of AOD pixels at 3-km resolution retrieved by the
Moderate Resolution Imaging Spectroradiometer (MODIS, https://ladsweb.nascom.nasa.gov/)
onboard the Aqua and Terra satellites (Levy et al., 2013). The MODIS 3-km AOD product
has been fully validated against ground-based sun photometers at both global (Remer et al.,
2013) and urban/suburban (Munchak et al., 2013) scales. Strada et al. (2015) used ground-
based AOD observations from the Aerosol Robotic Network (AERONET) near AMF sites to
validate the sampling technique of MODIS 3-km AOD product. They found high correlations
of 0.89-0.98 and regression slopes from 0.89 to 1.03 for daily AOD between AERONET and
MODIS at four AMF sites. For this study, the validation against ground-based AOD
observations was not possible because no AERONET stations exist near to the selected AMF
sites.

Every day, MODIS satellite sensors pass a specific region between 10:00 and 14:00 Local
Time (LT), leaving patchy signals around the AmeriFlux sites. Most of MODIS AOD data at
high latitudes are available only in boreal summer; as a result, we narrow our explorations of
the GPP-AOD relationships to the noontime (10:00-14:00 LT) from June to August. The
chosen noontime window limits the contributions that confounding factors such as low solar
angles and high diffuse fraction may have on the amount of diffuse PAR and plant
productivity (Niyogi et al., 2004). For each summer day, we select instantaneous MODIS 3-
km AOD pixels that are (a) located within a distance of 0.03° (about 3 km) from the targeted
AMF site and (b) "quasi-coincident" with AMF data, which are available each half-hour.
Because of the unavoidable temporal differences between MODIS overpass and AMF data
availability, we name this selection "quasi-coincident". A cloud mask applied to the MODIS
retrieval procedure conveniently filters out cloudy instants and should reduce the effect of
clouds in the scattering process. We calculate both the correlation and regression coefficients
between "quasi-coincident" GPP and AOD at the selected sites. Negative GPP is considered
as a missing value. To further reduce the influence of cloud cover, we discard instants (both
AMF and MODIS data) when precipitation is non-zero. In total, we select 65 pairs of GPP
and AOD at CA-Gro site and another 59 pairs at CA-Qfo site. The GPP-AOD sampling pairs
are much fewer than GPP-PAR, because we select instants when both instantaneous AOD
and GPP data are available. In addition, AOD is screened for clear instants to exclude the
impacts of clouds.

**2.2 Wildfire emissions**

Wildfire emissions used in climate modeling are calculated as the product of area burned,
fuel consumption, and emission factors. To predict area burned, we build stepwise
regressions for area burned in 12 boreal ecoregions (Yue et al., 2015). Observed area burned
aggregated from inter-agency fire reports is used as the predictand. Predictors are selected
from 44 (5×6+7×2) variables including five meteorological parameters (mean and maximum
temperature, relative humidity, precipitation, and geopotential height at 500 hPa) of six
different time intervals (winter, spring, summer, autumn, fire season (May-October), and the
whole year), as well as the mean and maximum values of 7 fire indexes from the CFWI
system during fire season. We consider the impacts of antecedent factors on current fire
activity by including all above variables at the same year and those in the previous two years,
making a total of 132 (44×3) factors. The final formats of regression are different among
ecoregions, depending on the selection of the factors that contribute the maximum observed
variance in predictand but remain the minimum collinearity among predictors. These
regression functions are then driven with output from 13 Coupled Model Intercomparison
Project Phase 3 (CMIP3) climate models under A1B scenario (Meehl et al., 2007) to predict
area burned at present day (1981-2000) and midcentury (2046-2065). In the A1B scenario,
$CO_2$ concentration is projected to 532 ppm by the year 2050, similar to the value of 541 ppm
in IPCC RCP8.5 scenario (van Vuuren et al., 2011) archived for the Coupled Model
Intercomparison Project Phase 5 (CMIP5).

We derive 1°×1° gridded area burned based on the prediction for each ecoregion following
the approach by Yue et al. (2015). Temporally, the annual area burned estimated with
regressions is first converted to monthly area burned using the mean seasonality for each
boreal ecoregion during 1980-2009. Spatially, large fires tend to burn in ecosystems where
historical fires are frequent because of favorable conditions (Keane et al., 2008). In each 1°×1°
grid square, we calculate the frequency of large fires (>1000 ha) during 1980-2009; these
fires account for about 85% of total area burned in boreal North America. We arbitrarily
attribute 85% of area burned within each ecoregion to a number of fires with fixed size of
1000 ha. We then allocate these large fires among the 1°×1° grid cells based on the observed
spatial probability of large fires. For example, if one grid box (named grid 'A') bears 1% of
large fires (>1000 ha) within an ecoregion at present day, the same grid will bear the same
possibility for large fires in the future. On the other hand, fuel availability limits reburning
and fire spread during the forest return interval, suggesting that current burning will decrease
the possibility of future fires in the same location. To consider such impact, we scale the
observed probabilities by the fraction remaining unburned in each grid box, and then use this
modified probability distribution to allocate large fires for the remaining months. For
example, if present-day fires have consumed 20% of the total area within the grid 'A', then
the possibility of large fire will be 0.8% (1%×0.8, instead of 1%) for this grid. Finally, we
disaggregate the remaining 15% of area burned into fires 10 ha in size, and randomly
distribute these fires across all grid boxes in the ecoregion. With this method, we derive the
gridded area burned for boreal North America by eliminating reburning issues. Sensitivity
tests show that specifying different area burned to the large fires (100 or 10 000 ha rather
than 1000 ha) yields < 1 % changes in predicted biomass burned, suggesting that this
approach is not sensitive to the presumed fire size in the allocation procedure.

Fuel consumption, the dry mass burned per fire area, is the product of fuel load and burning
severity. For fuel load in Alaska, we use 1-km inventory from the US Forest Service (USFS)
Fuel Characteristic Classification System (FCCS, McKenzie et al., 2007). For fuel load in
Canada, we use a 1-km fuel type map from the Canadian Fire Behavior Prediction (FBP)
system (Nadeau et al., 2005), combined with fuel-bed definition from the FCCS. Burning
severity, the fraction of fuel load burned by fires, is calculated with the USFS CONSUME
model 3.0 following the approach described in Val Martin et al. (2012). With both fuel load
and burning severity, we derive fuel consumption and further calculate biomass burned in
boreal North America with the predicted area burned. As in Amiro et al. (2009) and Yue et al.
(2015), we apply constant fuel load for both present day and midcentury because opposite
and uncertain factors influence future projections (Kurz et al., 2008; Heyder et al., 2011;
Friend et al., 2014; Knorr et al., 2016; Kim et al., 2017). Instead, we consider changes in
burning severity due to perturbations in fuel moisture as indicated by CFWI indexes (Yue et
al., 2015). On average, we estimate a 9% increase in fuel consumption over boreal North
America by the midcentury, because higher temperature and lower precipitation result in a
future with drier fuel load (Flannigan et al., 2016).

Fire emissions for a specific species are then estimated as the product between biomass
burned and the corresponding emission factor, which is adopted from measurements by
Andreae and Merlet (2001) except for $NO_x$. We use the average value of 1.6 g NO per Kg dry
mass burned (DM) from six studies as NOx emission factor, because the number of 3.0 g NO
per Kg DM reported in Andreae and Merlet (2001) is much higher than that of 1.1 g NO per
Kg DM from field observations (Alvarado et al., 2010). Based on projected area burned and
observation-based fuel consumption and emission factors, we derive fire emissions of $NO_x$,
carbon monoxide (CO), non-methane volatile organic compounds (NMVOCs, Alkenes and
Alkanes), $NH_3$, $SO_2$, black (BC) and organic carbon (OC) in the present day and midcentury.

**2.3 NASA ModelE2-YIBs model**


The NASA ModelE2-YIBs is an interactive climate-carbon-chemistry model, which couples
the chemistry-climate model NASA ModelE2 (Schmidt et al., 2014) and the YIBs vegetation
model (Yue and Unger, 2015). NASA ModelE2 is a general circulation model with
horizontal resolution of $2°×2.5°$ latitude by longitude and 40 vertical layers up to 0.1 hPa. It
dynamically simulates both the physical (emissions, transport, and deposition) and chemical
(production, conversion, and loss) processes of gas-phase chemistry ($NO_x$, $HO_x$, $O_x$, CO, $CH_4$,
and NMVOCs), aerosols (sulfate, nitrate, ammonium, BC, OC, dust, and sea salt), and their
interactions. In the model, oxidants influence the photochemical formation of secondary
aerosol species (e.g., sulfate, nitrate, and biogenic secondary organic aerosol), in turn,
aerosols alter photolysis rates and influence the online gas-phase chemistry. Size-dependent
optical parameters computed from Mie scattering, including extinction coefficient, single
scattering albedo, and asymmetry parameters, are applied for each aerosol type (Schmidt et
al., 2014). The model also considers interactions between climate and atmospheric
components. Simulated climate affects formation, transport, and deposition of atmospheric
components, in turn, both $O_3$ and aerosols influence climate by altering radiation, temperature,
precipitation, and other climatic variables. Both observation-based evaluations and multi-
model inter-comparisons indicate that ModelE2 demonstrates skill in simulating climatology
(Schmidt et al., 2014), soil moisture (Fig. S1), radiation (Wild et al., 2013), atmospheric
composition (Shindell et al., 2013b), and radiative effects (Shindell et al., 2013a).

YIBs is a process-based vegetation model that dynamically simulates changes in leaf area
index (LAI) through carbon assimilation, respiration, and allocation for prescribed PFTs.
Coupled photosynthesis-stomatal conductance is simulated with the Farquhar-Ball-Berry
scheme (Farquhar et al., 1980; Ball et al., 1987). Leaf-level photosynthesis is upscaled to
canopy level by separating diffuse and direct light for sunlit and shaded leaves (Spitters,
1986). Plant respiration considers thermal dependence as well as acclimation to temperature
(Atkin and Tjoelker, 2003). Soil respiration is calculated based on the carbon flows among 12
biogeochemical pools (Schaefer et al., 2008). Net carbon uptake is allocated among leaves,
stems, and roots to support leaf development and plant growth (Cox, 2001). The YIBs model
has been benchmarked against *in situ* GPP from 145 eddy covariance flux tower sites and
satellite retrievals of LAI and phenology (Yue and Unger, 2015). An interactive flux-based
$O_3$ damage scheme proposed by Sitch et al. (2007) is applied to quantify the photosynthetic
responses to ambient $O_3$ (Yue and Unger, 2014). For this scheme, $O_3$ damaging level is
dependent on excess $O_3$ stomatal flux within leaves, which is a function of ambient $O_3$
concentration, boundary layer resistance, and stomatal resistance. Reduction of
photosynthesis is calculated on the basis of plant functional types (PFTs), each of which
bears a range of low-to-high sensitivities to $O_3$ uptake.


**2.4 Simulations**

Using the NASA ModelE2-YIBs model, we perform 6 time-slice simulations, three for
present-day (2010s) and three for midcentury (2050s), with atmosphere-only configuration to
explore the impacts of fire emissions on NPP in boreal North America (Table 1). Simulations
F10CTRL and F50CTRL turn off all fire emissions as well as $O_3$ vegetation damage for the
2010s and 2050s, respectively. However, climatic feedbacks of aerosols from other sources
(both natural and anthropogenic) and related photosynthetic responses are included.
Simulations F10AERO and F50AERO consider the responses of plant productivity to
perturbations in radiation and meteorology caused by aerosols, including emissions from
wildfires and other sources, but do not include any $O_3$ vegetation damage. In contrast,
simulations F10O3 and F50O3 calculate offline $O_3$ damage based on the simulated $O_3$ from
all sources including fire emissions. For these simulations, reductions of GPP are calculated
twice with either low or high $O_3$ sensitivity. However, both of these GPP changes are not fed
back into the model to influence carbon allocation and tree growth. Plant respiration is
changing in response to meteorological perturbations, either due to climate change or aerosol
radiative effects. We assume no impact of $O_3$ damage to plant respiration and examine
vegetation NPP, the net carbon uptake by biosphere, for the current study. The difference
between AERO and CTRL runs isolates the impacts of fire aerosols on NPP, and the
difference between O3 and CTRL runs isolates $O_3$ vegetation damage caused by fire and non-
fire emission sources.

All simulations are conducted for 20 years and outputs for the last 15 years are used for
analyses. The simulations apply sea surface temperatures (SSTs) and sea ice distributions
from previous NASA GISS experiments under the IPCC RCP8.5 scenario (van Vuuren et al.,
2011). Decadal average monthly-varying SST and sea ice of 2006-2015 are used as boundary
conditions for present-day (2010s) runs while that of 2046-2055 are used for future (2050s)
runs. In the RCP8.5 scenario, global average SST increases by 0.62 °C while sea ice area
decreases by 13.8% at the midcentury compared to the present-day level. Decadal average
well-mixed greenhouse gas concentrations and anthropogenic emissions of short-lived
species, both at present day and midcentury, are adopted from the RCP8.5 scenario (Table 2).
The enhancement of $CO_2$ will affect climate (through longwave absorption) and ecosystem
productivity (through $CO_2$ fertilization), but not the fire activity and related emissions
directly. Natural emissions of soil and lightning $NO_x$, biogenic volatile organic compounds
(BVOC), dust, and sea salt are climate-sensitive and simulated interactively. The YIBs
vegetation model cannot simulates changes in PFT fractions. The RCP8.5 land cover change
dataset shows limited changes in land cover fractions between 2010s and 2050s (Oleson et al.,
2010). For example, relative to the 2010s, a maximum gain of 5% is predicted for grassland
in the 2050s, resulting from a 1% loss in deciduous forest and another 1% loss in needleleaf
forest over boreal North America. As a result, a land cover dataset derived from satellite
retrievals (Hansen et al., 2003) is applied as boundary conditions for both the 2010s and
2050s.

To evaluate the simulated GPP responses to changes in diffuse radiation, we perform site-
level simulations using standalone YIBs model, which is driven with observed hourly
meteorology (including temperature, relative humidity, surface pressure, wind speed, and soil
moisture) and both diffuse and direct PAR at sites CA-Gro and CA-Qfo. To isolate the
impact of individual aerosol-induced climatic perturbations on NPP, we perform 10
sensitivity experiments using the offline YIBs model driven with offline meteorology
simulated by ModelE2-YIBs model (Table 3). For example, the offline run Y10_CTRL is
driven with variables from the online simulation of F10CTRL (Table 1). The run Y10_TAS
adopts the same forcing as Y10_CTRL except for temperature, which is simulated by the
climate simulation of F10AERO. In this case, we quantify the NPP responses to individual
and/or combined climate feedback (mainly in temperature, radiation, and soil moisture) by
fire aerosols. Each offline run is conducted for 12 years and the last 10 years are used for
analyses.

**2.5 Observation datasets**

We use observations to evaluated GPP, AOD, and $O_3$ in boreal North America simulated by
ModelE2-YIBs. For GPP, we use a benchmark data product upscaled from FLUXNET eddy
covariance data using an ensemble of regression trees (Jung et al., 2009). For AOD
observations, we use satellite retrieval at 550 nm from Terra MODIS Level 3 data product.
For $O_3$, gridded datasets are not available. We use site-level observations from 81 U.S. sites
at the Clean Air Status and Trends Network (CASTNET, https://www.epa.gov/castnet) and
Canadian    sites    at    the    National    Air    Pollution    Surveillance    (NAPS,
http://www.ec.gc.ca/rnspa-naps/) program. All datasets are averaged over the 2008-2012
period to represent present-day climatological conditions. Gridded datasets are interpolated to
the same 2°×2.5° resolution as ModelE2-YIBs model.


**3  Results**

**3.1 Observed GPP-AOD relationships**

Positive correlations between GPP and diffuse PAR are found at the two boreal sites (Figs
2b-2c). The magnitude of diffuse PAR is similar for these sites, possibly because they are
located at similar latitudes (Fig. 2a). GPP values at CA-Gro are generally higher than that at
CA-Qfo, likely because deciduous broadleaf forest (DBF) has higher photosynthetic rates.
Consequently, the slope of regression between GPP and $PAR_{dif}$ is higher at CA-Gro than that
at CA-Qfo, suggesting that GPP of DBF (or MF) is more sensitive to changes in diffuse PAR
than that of ENF. We find almost zero correlation between GPP and $PAR_{dir}$ at the two sites
(Table 4), indicating that photosynthesis is in general light-saturated for sunlit leaves at these
sites during boreal summer noontime. As a result, modest reductions in direct light by
aerosols will not decrease GPP of the whole canopy.

With satellite-based AOD, we find positive correlations between GPP and AOD at both sites
(Figs 2d-2e). However, the slope of regression between GPP and AOD is lower (and not
significant) at CA-Gro compared with that at CA-Qfo, opposite to the GPP-PAR$_{dif}$
regressions. The cause of such discrepancy might be related to the limitation of data
availability. For the same reason, the GPP-AOD correlation is insignificant at CA-Gro site.
On average, GPP sensitivity (denoted as mean ± range) is estimated as 3.5 ± 1.1 μmol m$^{-2}$ s$^{-1}$
per unit AOD at lower latitudes of boreal regions in the summer.

**3.2 Model evaluations**

Simulated summer GPP shows high values in mid-western Canada (Alberta and
Saskatchewan) and the Southeast (Ontario) (Fig. 3a). Forest GPP at high latitudes is low
because of the cool weather and light limitation there. Simulated GPP reasonably captures the
spatial distribution with a high correlation coefficient of 0.77 ($p << 0.01$) and relatively small
biases within 20% of the data product. Simulated AOD reproduces the observed spatial
pattern including the high values in boreal forests (Fig. 3b). In contrast to the MODIS
observations, predicted AOD is relatively uniform over the West with a background value of
~0.1. This discrepancy explains the low correlation coefficient ($R = 0.25$, $p<0.01$) between
the model and MODIS data. The simulation fails to capture the high values in the west,
possibly due to a climate model underestimation of biogenic secondary organic aerosol,
which may be an important contribution over the western boreal forest. Simulated maximum
daily 8-hour average (MDA8) [$O_3$] shows low values in boreal North America and high
values in the western and eastern U.S. (Fig. 4a). This pattern is consistent with surface
observations (Fig. 4b), but the model overestimates the measured surface $O_3$ by 22%. The
Canadian measurement sites are located near the southern boundary, and as a result do not
represent the average state over the vast boreal region at higher latitudes.

With the Sitch et al. (2007) scheme, the YIBs model simulates reasonable GPP responses to
[$O_3$] in North America (Yue and Unger, 2014; Yue et al., 2016). Generally, damage to GPP
increases with the enhancement of ambient [$O_3$], but with varied sensitivities for different
plant species (see Fig. 6 of Yue and Unger (2014)). In responses to the same level of [$O_3$],
predicted $O_3$ damages are higher for deciduous trees than that for needleleaf trees, consistent
with observations from meta-analyses (Wittig et al., 2007). The model also reproduces

observed light responses of GPP to diffuse radiation in boreal regions. With the site-level simulations, we evaluate the modeled GPP-PAR$_{dif}$ relationships at the hourly (instead of half-hourly) time step during summer. For 1342 pairs of GPP and PAR$_{dif}$ at the site CA-Gro, the observed correlation coefficient is 0.42 and regression slope is 0.011, while the results for the simulation are 0.60 and 0.014, respectively. At the site CA-Qfo, the observations yield a correlation coefficient of 0.46 and regression slope of 0.007 for 1777 pairs of GPP and PAR$_{dif}$. The simulated correlation is 0.61 and the regression is 0.011 at the same site. The GPP sensitivity to PAR$_{dif}$ in the model is slightly higher than that of the available observations, likely because the latter are affected by additional non-meteorological abiotic factors. To remove the influences of compound factors other than radiation, we follow the approach of Mercado et al. (2009) to discriminate GPP responses to 'diffuse' and 'direct' components of PAR at the two sites (Fig. 5). The model successfully reproduces the observed GPP-to-PAR sensitivities. Increase in PAR boosts GPP, but the efficiency is much higher for diffuse light than that for direct light, suggesting that increase of diffuse radiation is a benefit for plant growth.

**3.3 Simulation of wildfire O$_3$ and aerosols**

During 1980-2009, wildfire is observed to burn $2.76 \times 10^6$ ha and 156.3 Tg DM every year over boreal North America. Similarly, the ensemble prediction with fire regression models estimates present-day area burned of $2.88 \times 10^6$ ha yr$^{-1}$ and biomass burned of 160.2 Tg DM yr$^{-1}$ (Yue et al., 2015). By the midcentury, area burned is projected to increase by 77% (to $5.10 \times 10^6$ ha yr$^{-1}$) in boreal North America, mainly because of the higher temperature in future fire seasons. Consequently, biomass burned increases by 93% (to 308.6 Tg DM yr$^{-1}$) because fuel consumption also increases by 9% on average in a drier climate (Yue et al., 2015). Enhanced fire emissions increase concentrations of surface O$_3$ and column AOD, especially over Alaska and central Canada (Fig. 6). The maximum centers of air pollutants are collocated for O$_3$ and AOD but with unproportional magnitudes, suggesting non-linear conversion among fire emission species as well as the interactions with natural emission sources (e.g., lightning/soil NO$_x$ and BVOC). On average, wildfire emissions contribute $7.1 \pm 3.1\%$ ($2.1 \pm 0.9$ ppbv) to surface O$_3$ and $25.7 \pm 2.4\%$ ($0.03 \pm 0.003$) to AOD in the summer over boreal North America in the present day. By midcentury, these ratios increase significantly to $12.8 \pm 2.8\%$ ($4.2 \pm 0.9$ ppbv) for O$_3$ and $36.7 \pm 2.0\%$ ($0.05 \pm 0.003$) for AOD.

**3.4 Simulation of fire pollution impacts on NPP**

Surface $O_3$, including both fire and non-fire emissions (Table 2), causes limited (1-2%) damages to summer GPP in boreal North America (Fig. 7). The most significant damage is predicted over eastern U.S., where observed $[O_3]$ is high over vast forest ecosystems (Fig. 4). In the western U.S., $[O_3]$ is also high but the $O_3$-induced GPP reduction is trivial because low stomatal conductance in the semi-arid ecosystems limits $O_3$ uptake there (Yue and Unger, 2014). Over boreal North America, dominant PFTs are ENF (accounting for 44% of total vegetation cover) and tundra (treated as shrubland, accounting for 41% of total vegetation cover). Both species have shown relatively high $O_3$ tolerance with a damaging threshold of 40 ppbv as calculated with Sitch's scheme (Yue and Unger, 2014). For boreal regions, the mean $[O_3]$ of 28 ppbv (Fig. 4a) is much lower than this damaging threshold, explaining why the excess $O_3$ stomatal flux (the flux causing damages) is low there (Fig. 8). Statistics in Yue et al. (2015) show that maximum daily 8-hour average (MDA8) $[O_3]$ with fire contributions can be higher than 40 ppbv in Alaska and Canada. However, such episodes appear at 95 percentile for present day and 90 percentile for midcentury, suggesting that $O_3$ vegetation damage is rare in boreal North America and fire-induced $O_3$ enhancement does not exacerbate such damages. Therefore, we do not consider $O_3$ damage effects further.

Fire aerosols cause significant perturbations in shortwave radiation at surface (Fig. 9). The direct light is largely attenuated especially over Alaska and central Canada, where fire aerosols are most abundant (Fig. 6). In contrast, diffuse light widely increases due to particle scattering. In the present day, the average reduction of 5.6 W m$^{-2}$ in the direct light component is in part offset by the enhancement of 2.6 W m$^{-2}$ in the diffuse light component, leading to a net reduction of 3.0 W m$^{-2}$ in solar radiation over boreal North America. By the midcentury, a stronger reduction of 9.5 W m$^{-2}$ in direct light is accompanied by an increase of 4.0 W m$^{-2}$ in diffuse light, resulting in a net reduction of 5.5 W m$^{-2}$ in solar radiation. Fire-induced BC aerosols strongly absorb solar radiation in the atmospheric column (Figs 10a-10b). On average, fire aerosols absorb 1.5 W m$^{-2}$ in the present day and 2.6 W m$^{-2}$ by the midcentury.

Atmospheric circulation patterns respond to the aerosol-induced radiative perturbations (Figs
10c-10d). Surface radiative cooling and atmospheric heating together increase air stability
and induce anomalous subsidence. In the present day, such descending motion is confined to
55-68°N, accompanied by a rising motion at 52-55°N (Fig. 10c). As a result, fire aerosols
induce surface warming at higher latitudes but cooling at lower latitudes in boreal regions
(Fig. 11a). Meanwhile, precipitation is inhibited by the subsidence in northwestern Canada
but is promoted by the rising motion in the Southwest (Fig. 11c). By the midcentury, the
range of subsidence expands southward to 42°N (Fig. 10d) due to strengthened atmospheric
heating (Fig. 10b). The downward convection of warm air offsets surface radiative cooling
(Fig. 9b), leading to a significant warming in the Southwest (Fig. 11b). The expanded
subsidence further inhibits precipitation in vast domain of Canada (Fig. 11d). Soil moisture is
closely related to rainfall and as a result exhibits dipole changes (drier north and wetter south)
in the present day (Fig. 11e) but widespread reductions (Fig. 11f) by the midcentury.

In response to the climatic effects of fire aerosols, boreal NPP shows distinct changes
between the present day and midcentury (Fig. 12). Such changes in NPP are a consequence of
changes in GPP and autotrophic respiration (Fig. S2). Variations in plant respiration resemble
those of GPP, because higher photosynthesis leads to faster leaf/tissue development, resulting
larger maintenance and growth respiration. In the 2010s, forest NPP increases by 5-15% in
Alaska and southern Canada, but decreases by 5-10% in northern and eastern Canada. This
pattern of NPP changes ($\Delta$NPP) is connected to the climatic effects of aerosols, especially
changes in soil moisture (Fig. 11). The correlation between $\Delta$NPP (Fig. 12a) and changes in
soil moisture (Fig. 11e) reaches $R = 0.56$ (n = 356), much higher than the values of $R = -0.11$
for temperature change (Fig. 11a) and $R = 0.22$ for precipitation change (Fig. 11c). At the
continental scale, the patchy responses of NPP offset each other. Since the dominant fraction
of carbon uptake occurs in southern Canada (Fig. 3a), where positive NPP change is
predicted (Fig. 12a), wildfire aerosols enhance the total NPP by 72 Tg C yr$^{-1}$ in the present
day (Table 5). In contrast, increased wildfire emissions in the 2050s inhibit precipitation (Fig.
11d) and decrease soil moisture in boreal North America (Fig. 11f), leading to widespread
NPP reductions and a total NPP loss of 118 Tg C yr$^{-1}$ (Fig. 12b, Table 5).


**4 Discussion**

**4.1 Roles of aerosol climatic feedback**


The contrasting sign of NPP responses in the present day and midcentury are closely related
to the aerosol-induced surface climatic feedback. Sensitivity experiments using offline YIBs
model (Table 3) allowed assessment of the impacts of individual changes in the major
meteorological drivers, including temperature, radiation (diffuse and direct), and soil
moisture (Table 5). The offline simulations driven with changes in all three variables yield
$\Delta$NPP of 126 Tg C yr$^{-1}$ for the 2010s and -97 Tg C yr$^{-1}$ for the 2050s. These values are
different from the online simulations, which predict $\Delta$NPP of 72 Tg C yr$^{-1}$ for the 2010s and -
118 Tg C yr$^{-1}$ for the 2050s. Missing of other aerosol climatic feedbacks in the offline model,
for example, changes in relative humidity, surface pressure, soil temperature, and turbulence
momentum, may cause such discrepancy between the online and offline simulations.
Seasonal analyses show that summertime $\Delta$NPP is 99 Tg C at present day and -95 Tg C at
midcentury, dominating the NPP changes all through the year, because both wildfire
emissions and ecosystem photosynthesis maximize in boreal summer.

Observations show that aerosols can promote plant photosynthesis through increasing diffuse
radiation (Niyogi et al., 2004; Cirino et al., 2014; Strada et al., 2015). Our analyses with
ground data also show positive correlations between GPP and PAR$_{dif}$ (Fig. 2 and Table 4),
and the model reproduces observed GPP responses to perturbations in direct and diffuse PAR
(Fig. 5). Wildfire aerosols enhance diffuse radiation by 2.6 W m$^{-2}$ (1.7%) at present day and
4.0 W m$^{-2}$ (2.3%) at midcentury in boreal North America (Fig. 9). With these changes,
simulated NPP increases by 8 Tg C yr$^{-1}$ at the 2010s and 14 Tg C yr$^{-1}$ at the 2050s (Table 5).
Near the two AmeriFlux sites (Fig. 2a), wildfires increase local AOD by 0.03 (Fig. 6c).
Meanwhile, we estimate that summer average (00:00-24:00) GPP increases by 0.04 $\mu$mol m$^{-2}$
s$^{-1}$ in the same region due to aerosol diffuse fertilization effects (DFE) based on the results of
(Y10_PAR – Y10_CTRL). This change suggests a simulated GPP sensitivity of 1.2 $\mu$mol m$^{-2}$
s$^{-1}$ (22%) per unit AOD. Observed GPP sensitivity to AOD at the two sites are 2.3 (19%) and
4.5 $\mu$mol m$^{-2}$ s$^{-1}$ (58%) per unit AOD, respectively (Figs 2d-2e). The absolute value of GPP
sensitivity from simulations is much smaller than that of observations, because the former is
for 24-h average while the latter is only for noontime (10:00-14:00). The relative change of
22% in YIBs model falls within the observed range of 19-58%.

The estimated NPP changes of 8 Tg C yr$^{-1}$ by the radiative effects of boreal fire aerosols are
much weaker than the enhancement of 78-156 Tg C yr$^{-1}$ by fires in Amazon basin (Rap et al.,
2015). There are at least two reasons for such a difference in the DFE between boreal and
Amazon fire aerosols. First, wildfire emissions and associated impacts on radiation are much
smaller in boreal regions. Wildfires in Alaska and Canada directly emit 68 Tg C yr$^{-1}$ at the
2010s, resulting in enhancement of summer AOD by 35% and diffuse radiation by 1.7%.
These boreal emissions are much smaller than the ~240 Tg C yr$^{-1}$ in Amazon basin (van der
Werf et al., 2010), where fires enhances regional PM2.5 concentrations by 85% and diffuse
radiation by 6.2% in dry seasons (Rap et al., 2015). Second, larger solar insolation in lower
latitudes allows stronger DFE for the same unit change of diffuse radiation. In our prediction,
most of NPP changes occur at high latitudes of boreal regions (Fig. 12), where total
insolation is not so abundant as that at the tropical areas. Consequently, decline of direct
radiation in boreal regions more likely converts the light availability of sunlit leaves from
light-saturation to light-limitation, offsetting the benefit from enhanced diffuse radiation for
shaded leaves. For this study, we do not find GPP reduction by the decline of direct light at
the two Ameriflux sites (Table 4), possibly because these sites are located at middle latitudes
(<50°N). In the future, more observations at higher latitudes (> 55°N) are required to explore
the sensitivity of GPP to AOD at the light-limited conditions.

Simulations have shown that absorbing aerosols can cause regional drought by increasing air
stability (Liu, 2005; Cook et al., 2009; Tosca et al., 2010). Our results confirm such tendency
but with varied range of hydrological responses depending on the magnitude of wildfire
emissions (Figs 11c-11f). Observations suggest that precipitation (and the associated soil
moisture) is the dominant driver of the changes in GPP over North America, especially for
the domain of cropland (Beer et al., 2010). Sensitivity experiments with offline YIBs model
show that changes in soil moisture account for 82.5% of ΔNPP at present day and 70.5% of
ΔNPP at midcentury (Table 5). These results suggest that aerosol-induced changes in soil
water availability, instead of temperature and radiation, dominantly contribute to the changes
of boreal NPP, consistent with observational and experimental results (Ma et al., 2012;
Girardin et al., 2016; Chen et al., 2017).

**4.2 Limitations and uncertainties**

In this study, we examine the interactions among climate change, fire activity, air pollution,
and ecosystem productivity. To reduce the complexity of the interactions, we focus on the
most likely dominant feedback and thus main chain of events: "climate → fire → pollution →
biosphere' (Fig. 1). However, our choice of feedback analysis does not mean that the
interplay of other processes is unimportant. For example, climate-induced changes in
vegetation cover/types can influence fire activity by alteration of fuel load, and air pollution
by BVOC emissions (climate → biosphere → fire/pollution). In addition, other feedbacks may
amplify ecosystem responses but are not considered. For example, the drought caused by fire
aerosols in the midcentury (Fig. 11) may help increase fire activity (fire → pollution →
climate → fire). Furthermore, we apply fixed SSTs in the climate simulations because reliable
ocean heat fluxes for the future world were not available. Many previous studies have
investigated regional aerosol-climate feedbacks without ocean responses. For example, Cook
et al. (2009) found that dust-climate-vegetation feedback promotes drought in U.S., with a
climate model driven by prescribed SSTs. Similarly, Liu (2005) found fire aerosols enhance
regional drought using a regional climate model, which even ignores the feedback between
local climate and large-scale circulation. While we do concede that our experimental design
is not a complete assessment of all known processes and feedbacks, within these limitations,
this study for the first time quantifies the indirect impacts of wildfire on long-range
ecosystem productivity under climate change.

We use the ensemble projected fire emissions from Yue et al. (2015). Area burned is
predicted based on the simulated meteorology from multiple climate models. Such an
approach may help reduce model uncertainties in climatic responses to $CO_2$ changes (Collins
et al., 2013; Kirtman et al., 2014), but cannot remove the possible biases in the selection of
climate scenarios and fire models. All predictions in Yue et al. (2015) are performed under
the IPCC A1B scenario. With two different scenarios, A2 of high emissions and B2 of low
emissions, Balshi et al. (2009) showed that future area burned in boreal North America
increases at a similar rate until the 2050s, after which area burned in A2 scenario increases
much faster than that in B2 scenario. On average, boreal area burned in Balshi et al. (2009)
increases by ~160% at 2051-2060 compared with 2001-2010, much higher than the change of
66% in Yue et al. (2015). In contrast, Amiro et al. (2009) predicted that boreal area burned at
the $2\times CO_2$ scenario increases only by 34% relative to the $1\times CO_2$ scenario. This ratio is only
half of the estimate in Yue et al. (2015), which compared results between periods with
$1.44\times CO_2$ and $1\times CO_2$. The discrepancies among these studies are more likely attributed to
the differences in fire models. Although both Amiro et al. (2009) and Yue et al. (2015)
developed fire-weather regressions in boreal ecoregions, the former study did not include
geopotential height at 500 hPa and surface relative humidity as predictors, which make
dominant contributions to area burned changes in the latter study. On the other hand, Balshi
et al. (2009) developed nonlinear regressions between area burned and climate at grid scale,
which helps retain extreme values at both the temporal and spatial domain. Compared to
previous estimates, Yue et al. (2015) predicted median increases in future fire emissions over
boreal North America.

We apply constant land cover and fuel load for both present day and midcentury, but we
estimate an increase in fuel consumption due to changes in fuel moisture. Future projection of
boreal fuel load is highly uncertain because of multiple contrasting influences. For example,
using a dynamic global vegetation model (DGVM) and an ensemble of climate change
projections, Heyder et al. (2011) predicted a large-scale dieback in boreal-temperate forests
due to increased heat and drought stress in the coming decades. On the contrary, projections
using DGVMs show a widespread increase in vegetation carbon under the global warming
scenario with $CO_2$ fertilization of photosynthesis (Friend et al., 2014; Knorr et al., 2016). In
addition, compound factors such as greenhouse gas mitigation (Kim et al., 2017), population
change (Knorr et al., 2016), pine beetle outbreak (Kurz et al., 2008), and fire management
(Doerr and Santin, 2016) may exert varied impacts on future vegetation and fuel load.
Although we apply constant fuel load, we consider changes of fuel moisture because warmer
climate states tend to dry fuel and increase fuel consumption (Flannigan et al., 2016). With
constant fuel load but climate-driven fuel moisture, we calculate a 9% increase in boreal fuel
consumption by the midcentury (Yue et al., 2015). Although such increment is higher than
the prediction of 2-5% by Amiro et al. (2009) for a doubled-$CO_2$ climate, the consumption-
induced uncertainty for fire emission is likely limited because changes in area burned are
much more profound.

Predicted surface [$O_3$] is much higher than observations over boreal North America (Fig. 4).
This bias does not affect main conclusions of this study, because predicted $O_3$ causes limited
damages to boreal GPP even with the overestimated [$O_3$] (Fig. 7). The result confirms that
fire-induced $O_3$ vegetation damage is negligible in boreal North America. For aerosols, the
model captures reasonable spatial pattern of AOD but with a background value of ~0.1
outside fire-prone regions, where the observed AOD is usually 0.1-0.2 (Fig. 3). This
discrepancy may be related to the insufficient representations of physical and chemical
processes in the model, but may also result from the retrieval biases in MODIS data due to
the poor surface conditions (Liu et al., 2005) and small AOD variations (Vachon et al., 2004)
at high latitudes.

Simulated aerosol climatic effects depend on radiative and physical processes implemented in
the climate model. We find that present-day boreal fire aerosols on average absorb 1.5 W m$^{-2}$
in the atmosphere (Fig. 10), which is much smaller than the value of 20.5 ± 9.3 W m$^{-2}$ for
fires in equatorial Asia (Tosca et al., 2010). This is because boreal fires enhance AOD only
by 0.03 while tropical fires increase AOD by ~0.4. Previous modeling studies showed that
fire plumes induce regional and downwind drought through enhanced atmospheric stability
(Feingold et al., 2005; Tosca et al., 2010; Liu et al., 2014). Most of these results were based
on the direct and/or semi-direct radiative effects of fire aerosols. Inclusion of the indirect
aerosol effect may further inhibit precipitation and amplify drought, but may also introduce
additional uncertainties for the simulations. The fire-drought interaction may promote fire
activity, especially in a warmer climate. Ignoring this interaction may underestimate future
area burned and the consequent emissions.

**4.3 Implications**

Inverse modeling studies have shown that the land ecosystems of boreal North America are
carbon neutral in the present day, with the estimated land-to-air carbon flux from -270 ± 130
Tg C yr$^{-1}$ to 300 ± 500 Tg C yr$^{-1}$ (Gurney et al., 2002; Rodenbeck et al., 2003; Baker et al.,
2006; Jacobson et al., 2007; Deng et al., 2014). Here, we reveal a missing land carbon source
due to future wildfire pollution, taking into account full coupling among fire activity, climate
change, air pollution, and the carbon cycle. Fire pollution aerosol increases boreal NPP by 72
Tg C yr$^{-1}$ in the present day, comparable to the direct carbon loss of 68 Tg C yr$^{-1}$ from
wildfire $CO_2$ emissions (product of biomass burned and $CO_2$ emission factors). By
midcentury, increasing fire emissions instead cause a NPP reduction of 118 Tg C yr$^{-1}$ due to
the amplified drought. Although NPP is not a direct indicator of the land carbon sink,
reduction of NPP is always accompanied with the decline of net ecosystem exchange (NEE)
and the enhanced carbon loss. In combination with the enhanced carbon emission of 130 Tg
C yr$^{-1}$, future boreal wildfire presents an increasing threat to the regional carbon balance and
global warming mitigation. Furthermore, the NPP reductions are mostly located in southern
Canada, where cropland is the dominant ecosystem, newly exposing the future wildfire-
related air pollution risk to food production.

Our analyses of fire pollution effects on boreal North American productivity may not be
representative for other boreal ecosystems and/or on the global scale. There is substantial
variability in plant species, topography, and climatology across different boreal regions. Such
differences indicate distinct GPP sensitivities as well as fire characteristics. At lower latitudes,
where anthropogenic pollution emissions are more abundant, ambient ozone concentrations
may have exceeded damaging thresholds for most plant species. In those regions, additional
ozone from a fire plume may cause more profound impacts on photosynthesis than our
estimate for boreal North America. For example, Amazonian fire is predicted to reduce forest
NPP by 230 Tg C yr$^{-1}$ through the generation of surface ozone (Pacifico et al., 2015).
Meanwhile, solar radiation is more abundant at lower latitudes, indicating more efficient
increases in photosynthesis through aerosol DFE because the sunlit leaves receive saturated
direct light in those regions. As shown in Beer et al. (2010), partial correlations between GPP
and solar radiation are positive in boreal regions but negative over the subtropics/tropics,
suggesting that light extinction by fire aerosols has contrasting impacts on plant
photosynthesis in the high versus low latitudes. Further simulations and analyses are required
to understand the net impacts of ozone and aerosols from biomass burning on the global
carbon cycle.


*Acknowledgements.* Xu Yue acknowledges funding support from the National Key Research
and Development Program of China (Grant No. 2017YFA0603802), the National Basic
Research Program of China (973 program, Grant No. 2014CB441202) and the "Thousand
Youth Talents Plan". Nadine Unger acknowledges funding support from The University of
Exeter.



**Table 1.** Online simulations with ModelE2-YIBs climate model [a]

| Simulations | SST | [CO$_2$] | Emissions | Fires | O$_3$ effect | Aerosol effect |
|---|---|---|---|---|---|---|
| F10O3 | 2010s | 2010s | 2010s | 2010s | Yes | No |
| F10AERO | 2010s | 2010s | 2010s | 2010s | No | Yes |
| F10CTRL | 2010s | 2010s | 2010s | No | No | Yes |
| F50O3 | 2050s | 2050s | 2050s | 2050s | Yes | No |
| F50AERO | 2050s | 2050s | 2050s | 2050s | No | Yes |
| F50CTRL | 2050s | 2050s | 2050s | No | No | Yes |


[a] Values of SST, [CO$_2$], and emissions are adopted from RCP8.5 scenario, with the average
of 2006-2015 for the 2010s and that of 2046-2055 for the 2050s. For fire emissions, values at
the 2010s are predicted based on meteorology for 1981-2000 and those at the 2050s are for

2046-2065.



**Table 2.** Emissions from wildfires and non-fire sources over boreal North America

| Species | Fire emissions (Tg yr$^{-1}$) | | Non-fire emissions (Tg yr$^{-1}$) | |
|---|---|---|---|---|
| | 2010s | 2050s | 2010s | 2050s |
| NO$_x$ [a] | 0.39 | 0.74 | 2.43 | 2.08 |
| CO | 15.7 | 28.8 | 5.9 | 4.0 |
| SO$_2$ [a] | 0.12 | 0.22 | 1.95 | 1.28 |
| NH$_3$ | 0.22 | 0.40 | 0.80 | 1.15 |
| BC | 0.08 | 0.16 | 0.03 | 0.01 |
| OC | 1.10 | 2.04 | 0.04 | 0.02 |
| NMVOC | 0.39 | 1.34 | 0.49 | 0.30 |
| BVOC [b] | N/A | N/A | 15.3 | 15.1 |

[a] Natural emissions are included for NO$_x$ (lightning and soil) and SO$_2$ (volcano).
[b] ModelE2-YIBs calculates BVOC emissions using photosynthesis-dependent scheme
implemented by Unger et al. (2013).

**Table 3.** Simulations with YIBs vegetation model driven by offline meteorology from
ModelE2-YIBs climate model

| Simulations | Base forcing | Temperature | PAR | Soil moisture |
|---|---|---|---|---|
| Y10_CTRL | F10CTRL | | | |
| Y10_ALL | F10CTRL | F10AERO | F10AERO | F10AERO |
| Y10_TAS | F10CTRL | F10AERO | | |
| Y10_PAR | F10CTRL | | F10AERO | |
| Y10_SLM | F10CTRL | | | F10AERO |
| Y50_CTRL | F50CTRL | | | |
| Y50_ALL | F50CTRL | F50AERO | F50AERO | F50AERO |
| Y50_TAS | F50CTRL | F50AERO | | |
| Y50_PAR | F50CTRL | | F50AERO | |
| Y50_SLM | F50CTRL | | | F50AERO |




**Table 4.** Pearson's correlation coefficients for GPP-PAR and GPP-AOD relationships at
Ameriflux (AMF) sites [a]

| Site | Period [b] | Pearson's $R$ | | | | | |
| --- | --- | --- | --- | --- | --- | --- | --- |
| | | GPP-PAR | GPP-PAR$_{dir}$ | GPP-PAR$_{dif}$ | GPP-AOD | AOD-PAR$_{dif}$ | AOD-PAR$_{dir}$ |
| CA-Gro | 2004-2013 | **0.19** (2432) | -0.01 (2432) | **0.42** (2432) | 0.15 (65) | **0.60** (65) | **-0.52** (65) |
| CA-Qfo | 2003-2014 | **0.16** (3201) | **-0.04** (3201) | **0.45** (3201) | **0.36** (59) | **0.91** (34) | **-0.80** (34) |


[a] Both GPP and PAR (direct PAR$_{dir}$ and diffuse PAR$_{dif}$) data are adopted from site-level AMF
measurements. AOD data are adopted from instantaneous MODIS Aqua and Terra 3-km
retrievals. Correlations are calculated for quasi-coincident AMF and MODIS data over
summer noontime (June-August, 10:00-14:00 Local Time). The sampling number for each
correlation is denoted in brackets. Significant ($p<0.05$) correlation coefficients are bolded.
[b] For CA-Gro site, diffuse PAR observations of 2005-2009 have been discarded because of
poor calibration, as documented on the AMF website.





**Table 5.** Changes in NPP (Tg C yr$^{-1}$) caused by composite and individual climatic effects of fire aerosols


|  | 2010s | 2050s |
| --- | --- | --- |
| Online [a] | 72 | -118 |
| Offline total [b] | 126 | -97 |
| Temperature | 11 | -22 |
| Radiation | 8 | 14 |
| Soil moisture | 104 | -86 |


[a] Online results are calculated using the ModelE2-YIBs model with (F10AERO – F10CTRL) for the 2010s and (F50AERO – F50CTRL) for the 2050s.

[b] Offline results are calculated with the YIBs model driven with individual or combined changes in temperature, radiation, and soil moisture.

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

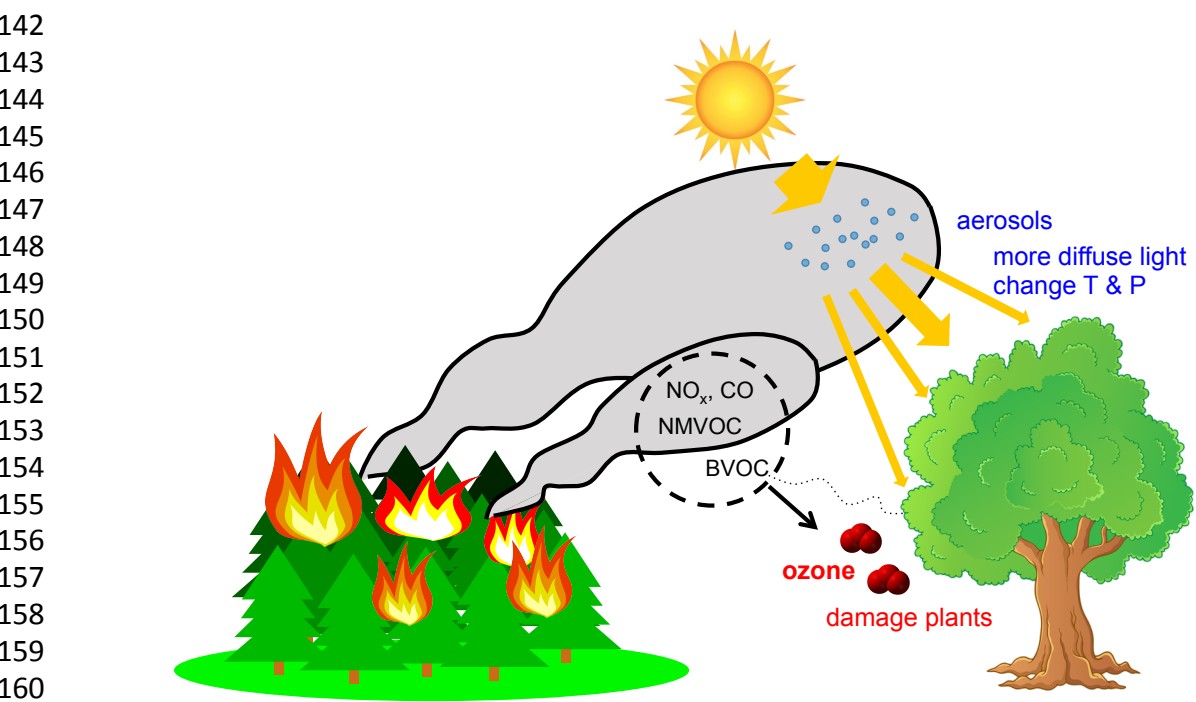

**Figure 1.** Illustration of atmospheric chemistry and physics, and biospheric processes investigated in the study. Carbonaceous aerosols from fire plumes increase diffuse light and change temperature and precipitation, influencing vegetation photosynthesis. Ozone generated photochemically from fire-emitted precursors (NOx, CO, and non-methane volatile organic compound (NMVOC)) and associated BVOC changes causes direct damage to plant photosynthesis.


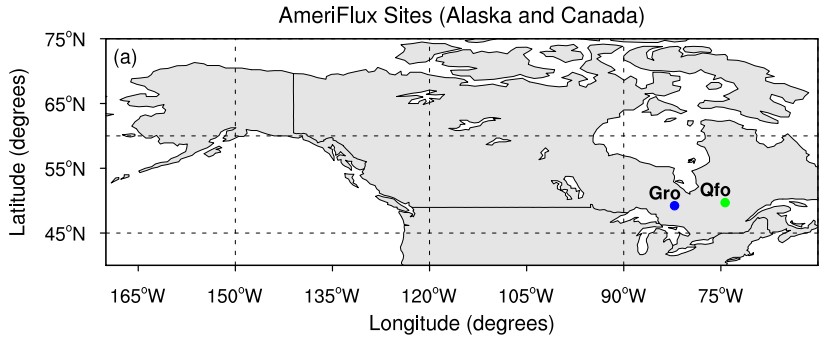

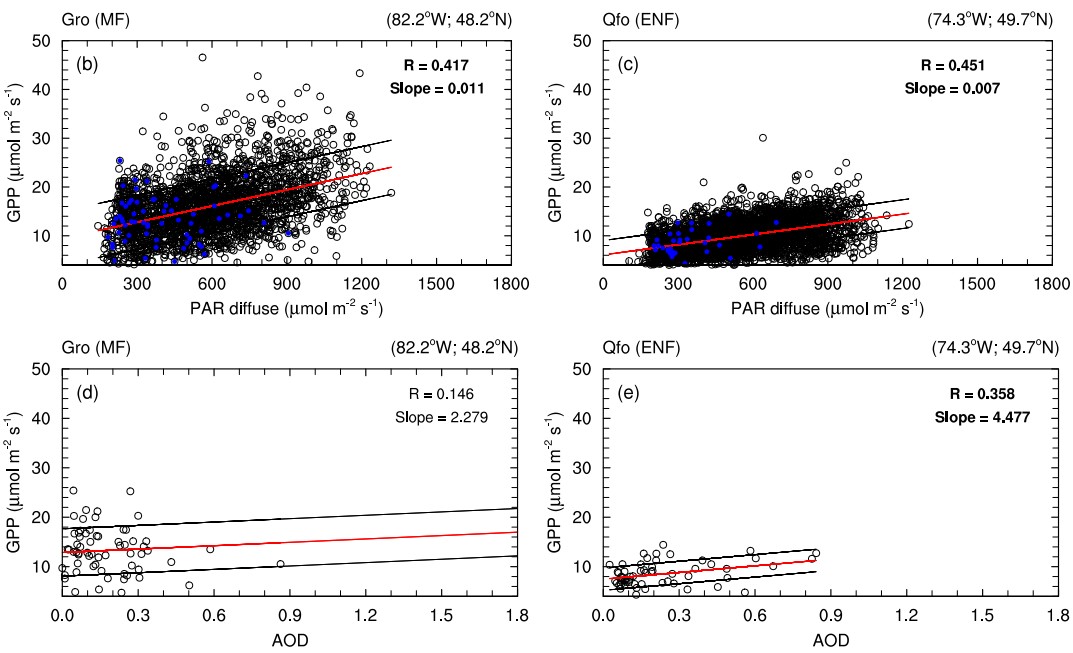

**Figure 2.** Relationships between (b, c) GPP and diffuse PAR and (d, e) GPP and MODIS
AOD at (a) two boreal sites: Groundhog River (Gro) and Quebec Mature Boreal Forest Site
(Qfo). The two sites are from the AmeriFlux network in Canada and are dominated by mixed
forest (MF at Gro) and evergreen needleleaf forest (ENF at Qfo) (Table 1). Data cover
summer days (June-August). AmeriFlux diffuse PAR and GPP (in µmol m$^{-2}$ s$^{-1}$) are half-
hourly observations (10:00-14:00 LT). Instantaneous MODIS Aqua and Terra 3-km AOD are
selected in a time span centered on AmeriFlux record time. For each plot: the red line
indicates the regression line, black lines depict the 1- $\sigma$ interval; the regression slope and
correlation coefficient are both included for each site (in bold if statistically significant at 95%
confidence level). Blue dots in (b, c) show instants when MODIS Aqua and Terra 3-km
AODs overlap AmeriFlux data.


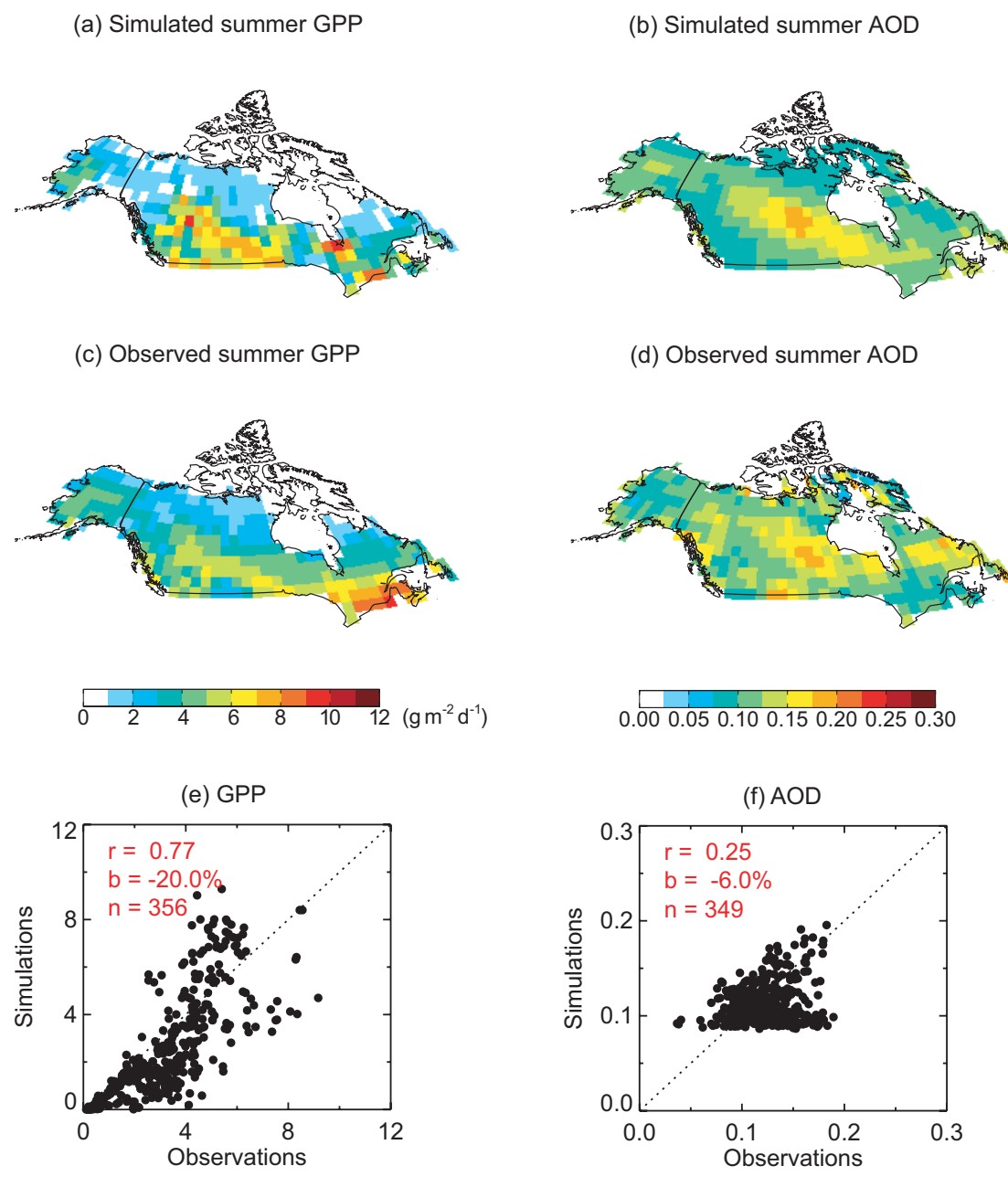


**Figure 3.** Evaluation of simulated summer (a) GPP and (b) AOD at 550 nm with (c, d) observations. Simulation results are from F10AERO (Table 1). Each point on the (e, f) scatter plot represents one grid square in boreal North America. The number of points (n), correlation coefficient (r), and relative bias (b) for the evaluation are presented on the plot.

1195

1196

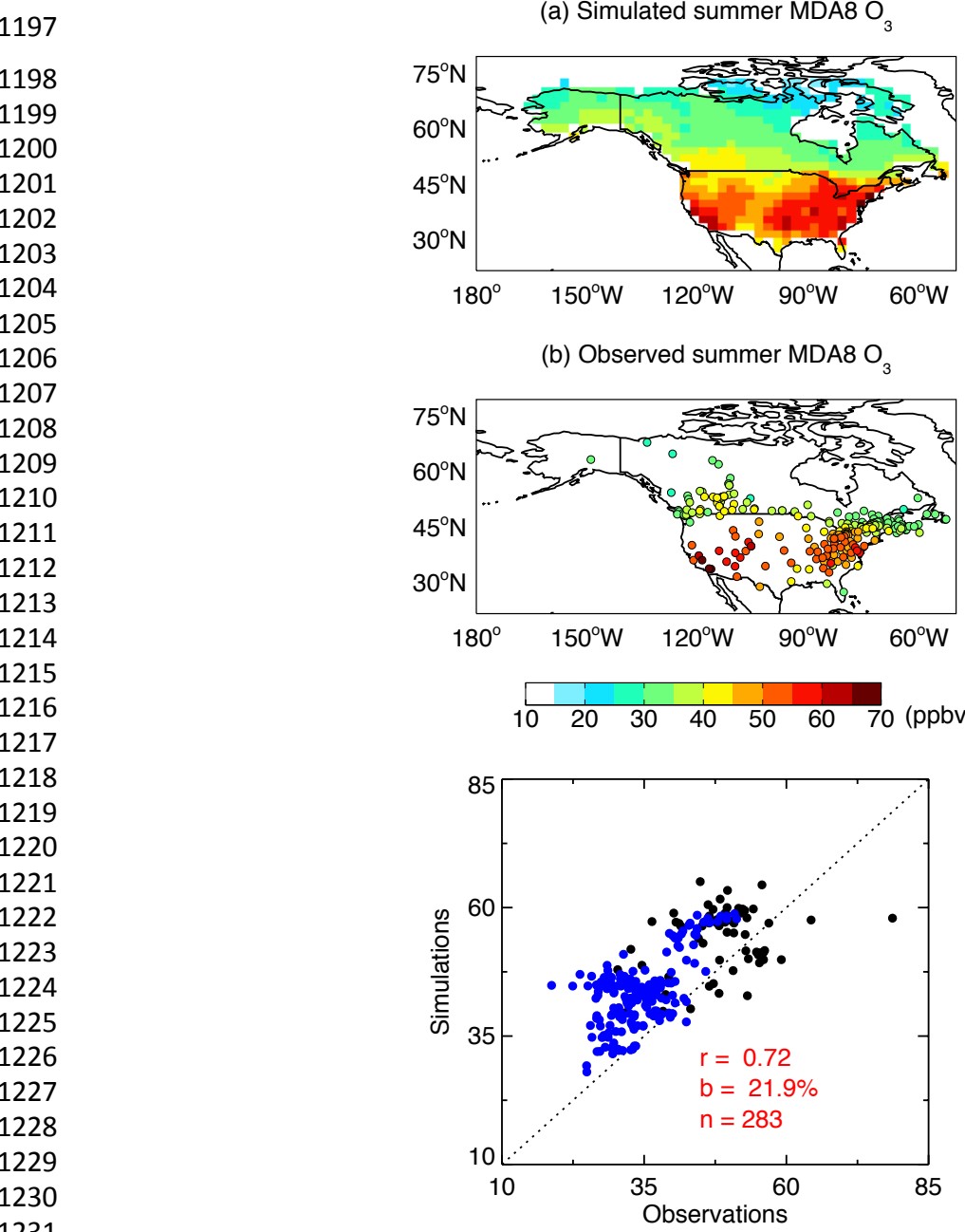

**Figure 4.** Evaluation of simulated summer surface maximum daily 8-hour average [$O_3$] with observations for 2008-2012. Observations are collected from 81 U.S. sites at the Clean Air Status and Trends Network (CASTNET) and 202 Canadian sites at the National Air Pollution Surveillance (NAPS) program. The number of points (n), correlation coefficient (r), and mean bias (b) for the evaluation are presented on the plot. Values over Canada and Alaska are denoted with blue points.


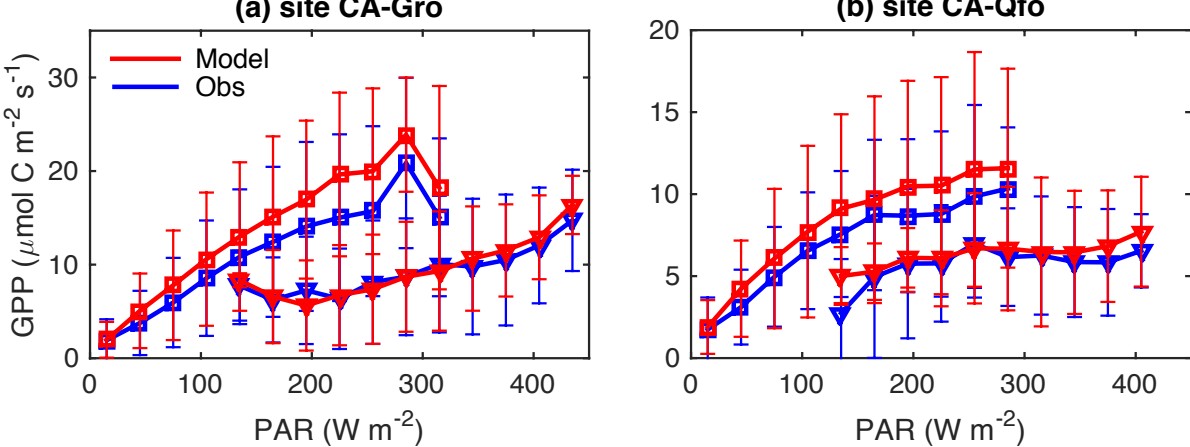

**Figure 5.** Observed (blue) and simulated (red) response of GPP to diffuse (square) and direct
(triangle) PAR at boreal sites (a) CA-Gro (2004-2013) and (b) CA-Qfo (2004-2010).
Observations and simulations are split into 'diffuse' and 'direct' conditions if the diffuse
fraction is >0.8 and < 0.2, respectively. Data points are then averaged over PAR bins of 30 W
$m^{-2}$ with error bars indicating one standard deviation of GPP for each bin.


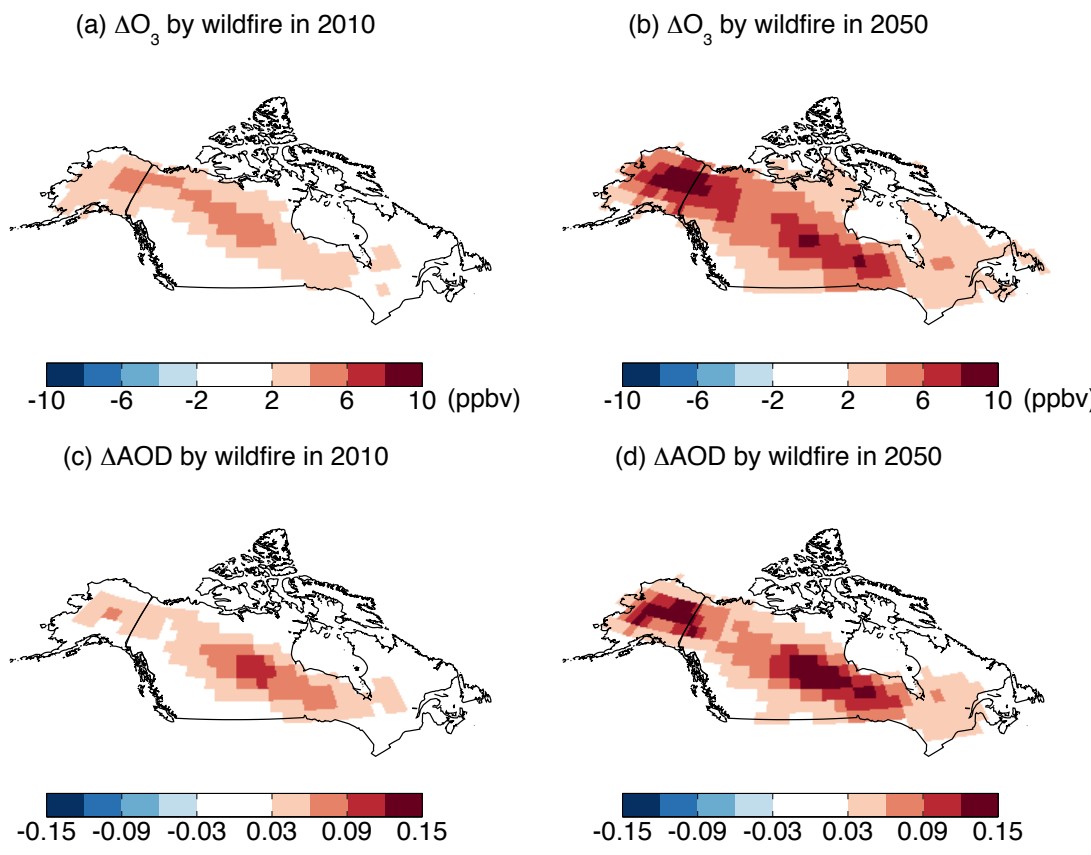


**Figure 6.** Changes in summer (a, b) [$O_3$] and (c, d) AOD at 550 nm induced by wildfire
emissions in (a, c) the 2010s and (b, d) the 2050s over boreal North America. Only
significant changes ($p<0.05$) are shown.


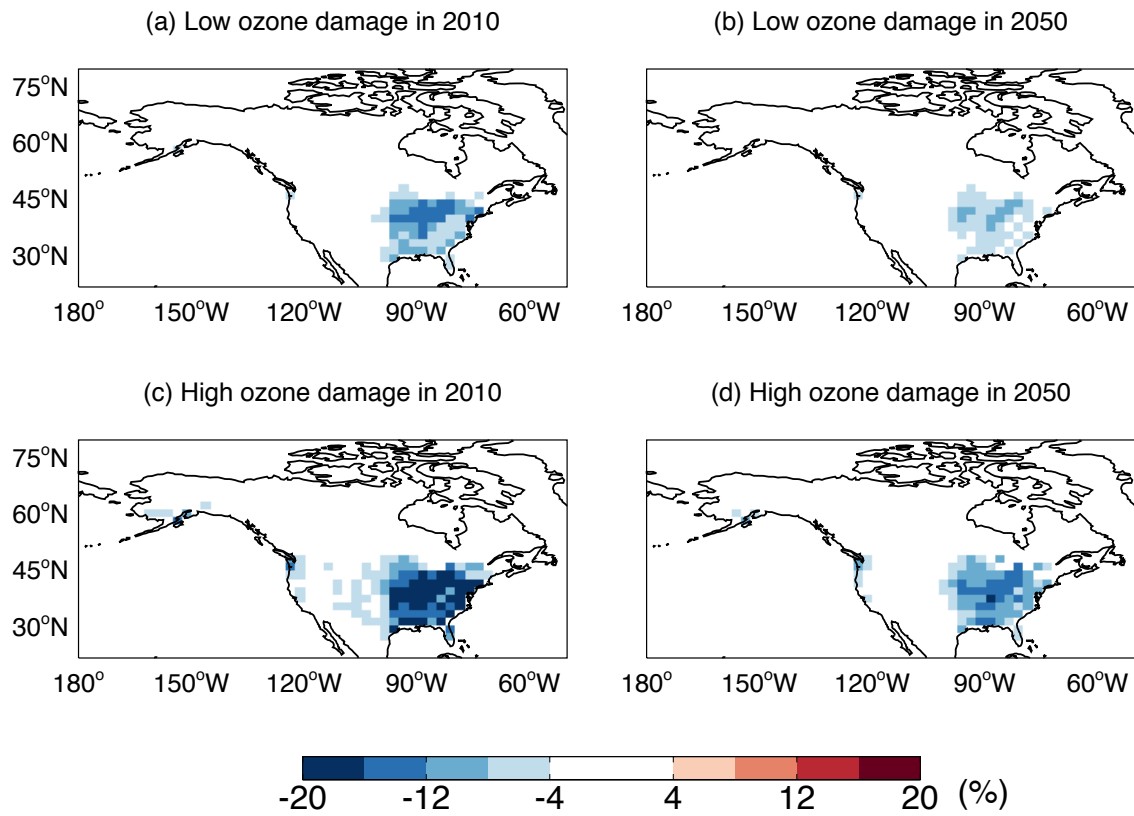

**Figure 7.** Simulated $O_3$ damages to summer GPP in North America. Results shown are from simulations with (a, b) low and (c, d) high $O_3$ sensitivities for (a, c) 2010 and (b, d) 2050. Simulated $[O_3]$ includes contributions from both wildfire and non-fire emissions. Results for 2010 are derived as (F10O3/F10CTRL-1)×100%. Results for 2050 are derived as (F50O3/F50CTRL-1)×100%.





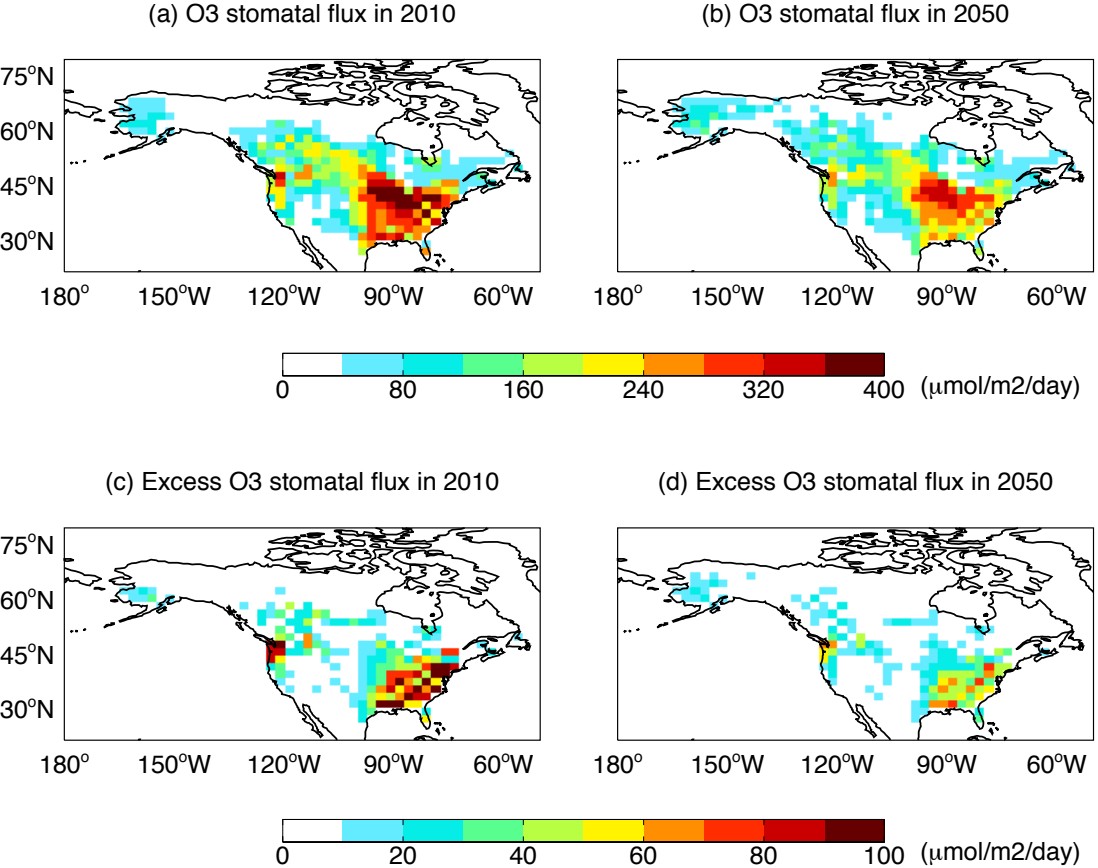

**Figure 8.** Simulated summertime O₃ stomatal fluxes in boreal North America. Results shown
are the (a, b) mean and (c, d) excess flux at (a, c) 2010 and (b, d) 2050. Simulated [O$_3$]
includes contributions from both wildfire and non-fire emissions. Excess O$_3$ stomatal flux is
calculated as the difference between the stomatal flux and a PFT-specific threshold as defined
in Sitch et al. (2007).


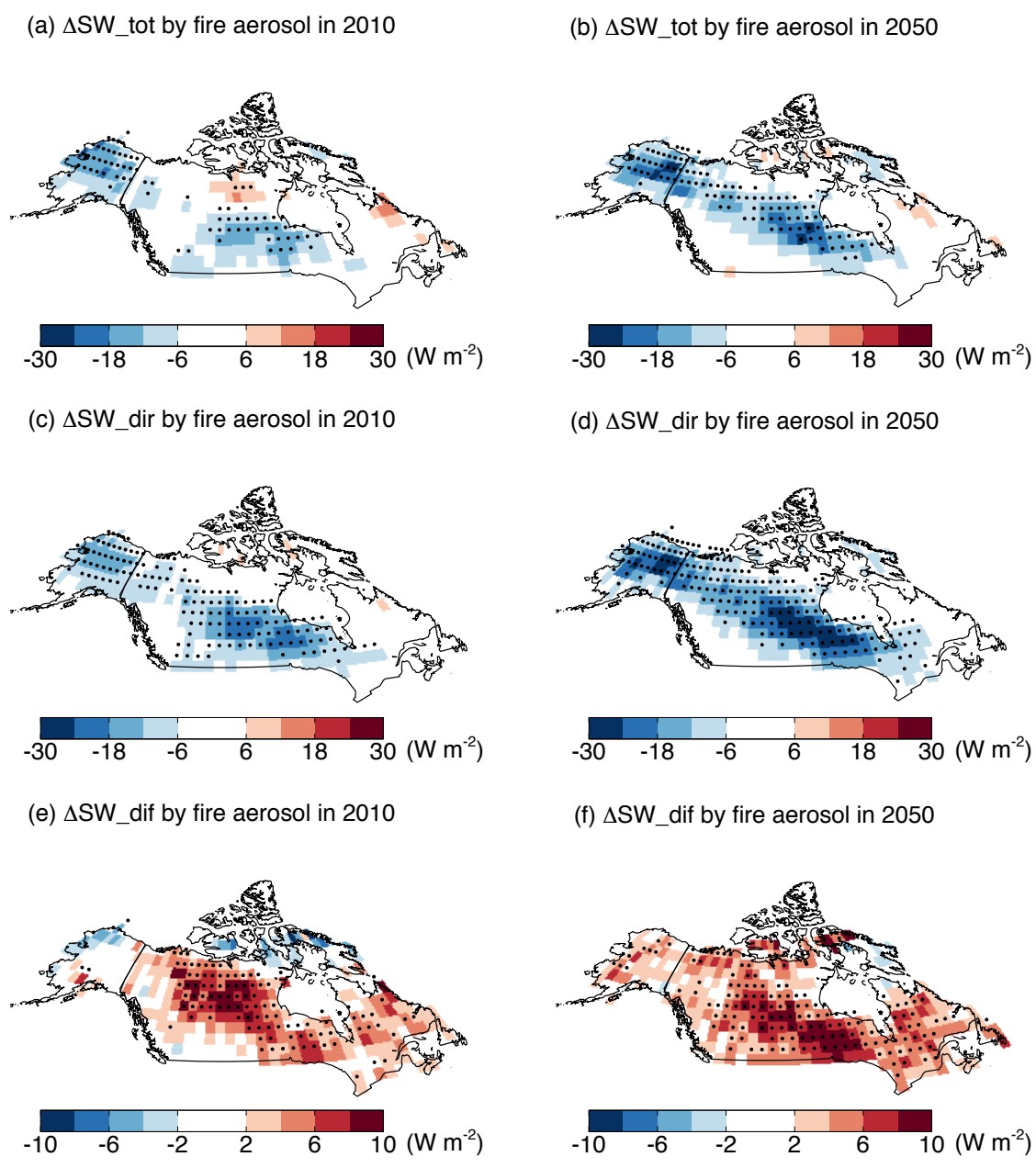

(a) ΔSW_tot by fire aerosol in 2010     (b) ΔSW_tot by fire aerosol in 2050

(c) ΔSW_dir by fire aerosol in 2010     (d) ΔSW_dir by fire aerosol in 2050

(e) ΔSW_dif by fire aerosol in 2010     (f) ΔSW_dif by fire aerosol in 2050

**Figure 9.** Changes in surface radiative fluxes induced by wildfire aerosols in boreal North
America. Results shown are for the changes in summertime (June-August) (a, b) total, (c, d)
direct, and (e, f) diffuse solar radiation at surface caused by aerosols from wildfire emissions
at (a, c, e) present day and (b, d, f) midcentury. Significant changes ($p<0.05$) are marked with
black dots. Results for 2010 are calculated as (F10AERO - F10CTRL). Results for 2050 are
calculated as (F50AERO - F50CTRL).

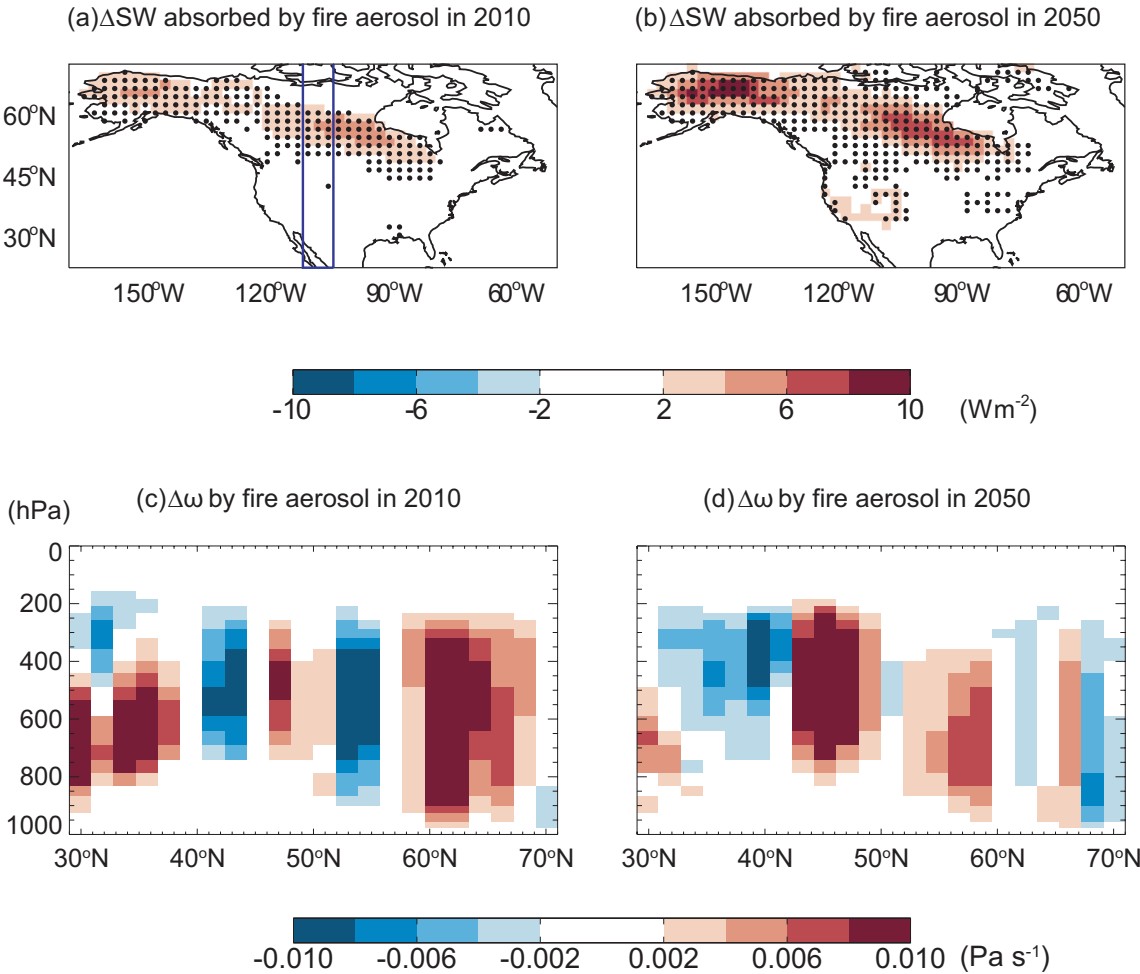

**Figure 10.** Predicted (a, b) absorption of shortwave radiation and (c, d) perturbations in
vertical velocity by wildfire aerosols at (a, c) present day and (b, d) midcentury. The
absorption of shortwave radiation is calculated as the differences of radiative perturbations
between top of atmosphere and surface. Vertical velocity is calculated as the longitudinal
average between 105°W and 112.5°W (two blue lines in a). Positive (negative) values
indicate descending (rising) motion. Results for the 2010s are calculated as (F10AERO -
F10CTRL). Results for the 2050s are calculated as (F50AERO - F50CTRL). Significant
changes ($p < 0.05$) in (a, b) are indicated as black points.

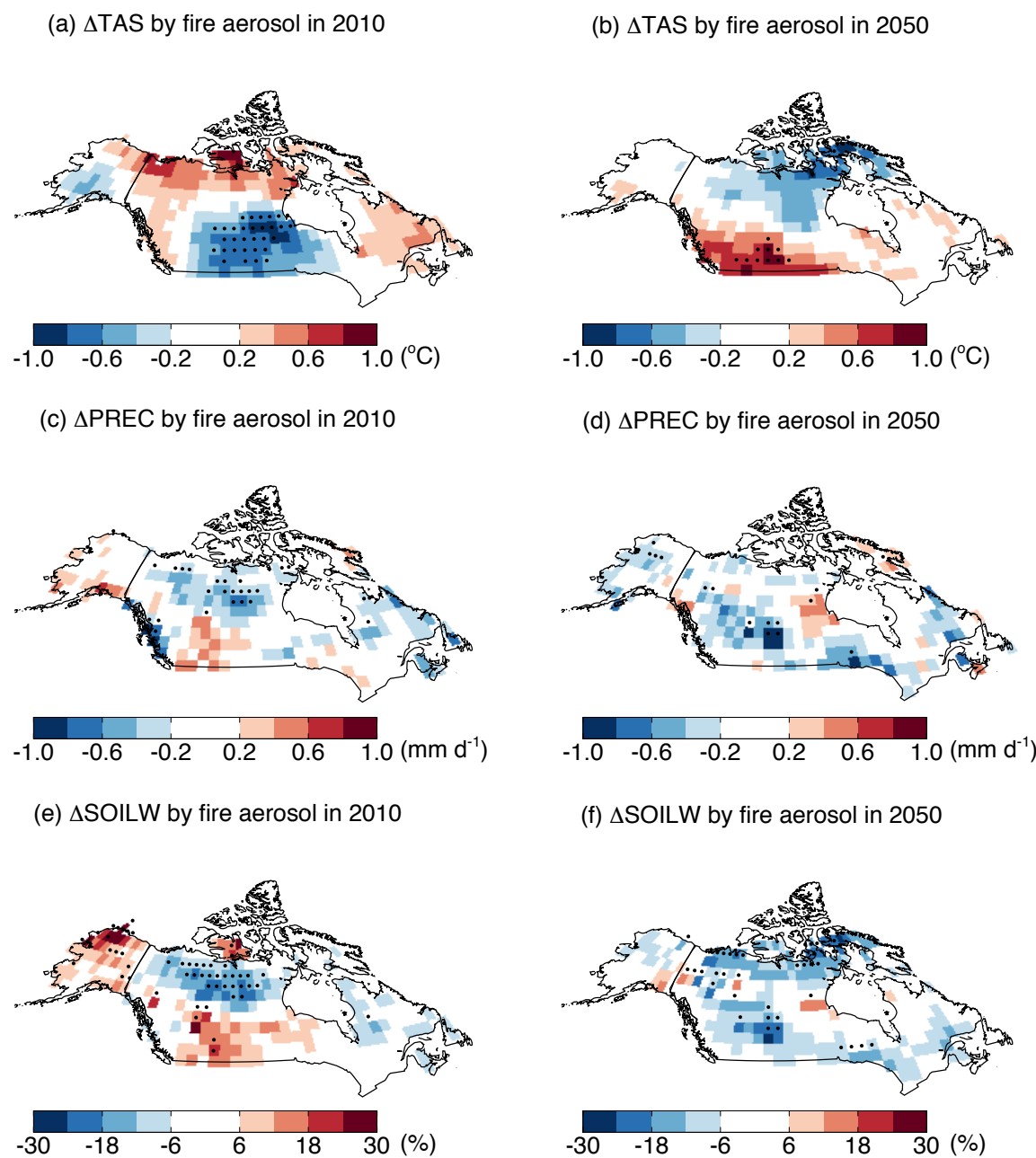

**Figure 11.** Predicted changes in summertime (a, b) surface air temperature, (c, d) precipitation, and (e, f) soil water content at surface caused by aerosols from wildfire emissions at (a, c, e) present day and (b, d, f) midcentury. Results for temperature and precipitation are shown as absolute changes. Results for soil water are shown as relative changes. Results for the 2010s are calculated as (F10AERO - F10CTRL). Results for the 2050s are calculated as (F50AERO - F50CTRL). Significant changes ($p<0.05$) are marked with black dots.

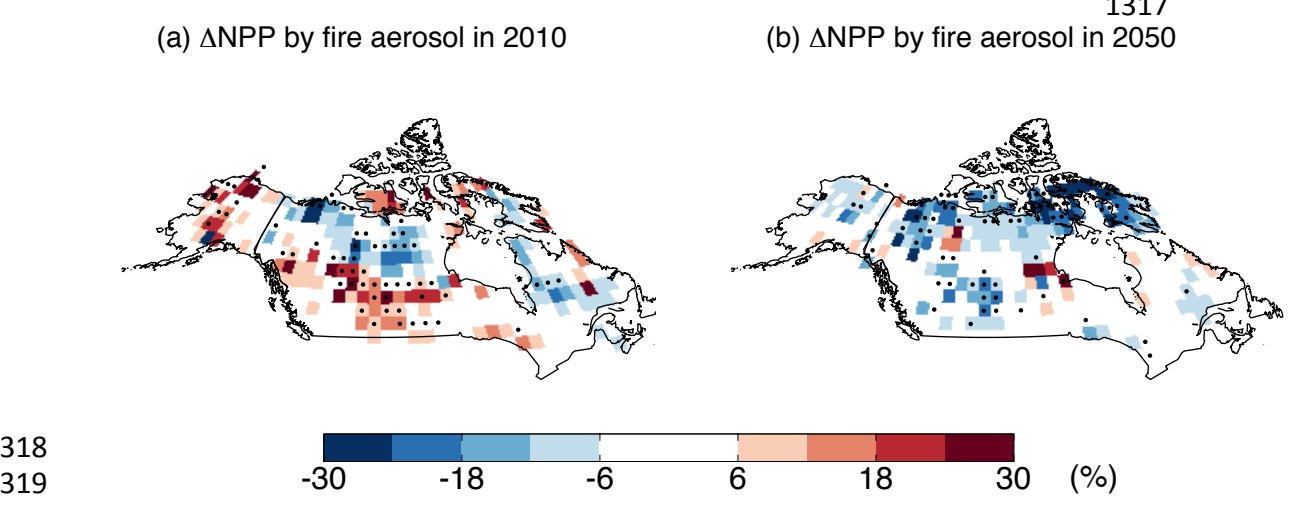

(a) ΔNPP by fire aerosol in 2010   (b) ΔNPP by fire aerosol in 2050

-30   -18   -6   6   18   30   (%)

**Figure 12.** Predicted percentage changes in summer NPP caused by wildfire aerosols at (a) present day and (b) midcentury. Results for the 2010s are calculated as (F10AERO/F10CTRL - 1) × 100%. Results for the 2050s are calculated as (F50AERO/F50CTRL - 1)×100%. Significant changes ($p<0.05$) are marked with black dots.