# Peer review of "Future inhibition of ecosystem productivity by increasing wildfire pollution"

_Atmospheric Chemistry and Physics, 2017_

## Referee Comment (RC1) · Anonymous Referee #1 · 24 Jun 2017

The manuscript is unusual in that it considers the indirect effect of wildfires on the boreal carbon balance via emissions of atmospheric pollutants. The results are novel and the simulated effect is surprisingly large, which makes the results interesting for ACP. There is some validation of results against observations, and some evaluation of effect strength directly using site-based observations. This strengthens the paper, which otherwise relies on a very complex modelling system. I consider the subject material to be fully within scope for ACP. However, the title does not correspond to the above assessment but sets different priorities. Possibly it reflects the original idea for the manuscript on ecosystem health but the focus has changed due to the negative results regarding ozone pollution. The manuscript therefore seriously lacks focus.

Major comments:

[Figure]

I can see at least the following scientific questions either being addressed, or requiring attention:

1) Do wildfires affect ecosystem health in boreal environments beyond its direct impact through higher ozone concentrations, i.e. far away from the fire or long after the fire has ended?

2) How can the effect be quantified, i.e. is NPP a valid proxy?

3) How do the direct and indirect effects compare?

4) Will the strength of this effect change in the future?

5) How do wildfires affect the carbon balance of boreal environments indirectly through atmospheric pollution away from the burned area?

6) How does this effect compare to the direct effect on the carbon cycle in the burned area?

7) How will this change in the future?

8) Are the results representative of all boreal regions?

Questions 1-4 correspond to the title, but Questions 5-8 to the actual focus of the paper (but still not all of them are being answered).

In order to become publishable, either the title needs to be changed to reflect the true focus of the paper, or the focus of the paper needs to be changed and much more detail on ecosystem health effects need to be included. The latter is probably beyond scope, so the best way forward must be the former. In that case, however, more depth is required regarding the carbon cycle, as NPP is only one of many components, and all of Questions 5-8 need to be answered. If the impact on the carbon cycle were to be the focus, then the title would have to be adapted and the manuscript would have to include more discussion that puts the results into the perspective of the regional and global carbon cycle. Some of it is there, but not enough to give the reader a sufficiently

good feel for how important this really is. So if the focus is to be on the carbon cycle, more results need to be included or a more detailed and in-depth discussion is needed. Or, what is also possible, restrict the paper to impacts on NPP alone. After all, you also include GPP, and that is already a step that involves changes in plant respiration, which also need to be projected. What happens here?

Finally, the result must be backed up more by measurements. The main effect is surprising, but it will be crucial that there is a thorough evaluation of how the model simulates the impact of changes in diffuse and direct light on GPP, as opposed to the measurements.

The third possibility would be to simply focus on the effect of atmospheric pollution from wildfires on GPP (not NPP) in boreal North America (and change the title accordingly).

Another major comment: the different chains of events discussed here are enormously complicated and the effect is very indirect. I suggest the authors show this in a suitable graphic. We have changes in climate affecting fire weather, but also affecting vegetation composition and fuel load. In addition we have changes in land use, in particular forestry and fire management (See Fig. 3 in Doerr and Satin showing for the U.S. increasing burned area, fewer fires, and an enormous rise in fire suppression costs). Both impact burned area and fire emissions. But then we also have atmospheric circulation patterns which are influenced by all sorts of things, among them greenhouse gas concentrations and aerosol load, some of it from boreal forest fires. And all of these together influence boreal forest NPP which in turn impacts the regional and global carbon cycle. Given this enormously complex web of causes and effects, I am not sure what we really learn here. It us up to the authors to clarify and give us a clear picture of what this paper is really about. Do that, I suggest considering the main questions and sub-questions as above, and then re-structuring the paper in order to answer them all in a systematic way. Much of it is there, but the information is too scattered.

Increases in boreal wildfire activity: this manuscript builds heavily on Yue et al. (2015),

which in turn builds heavily on Yue et al. (2013). This compartmentalisation of research is necessary given the said complexity of the subject. However, the foundations and basic assumptions on which the story rests here get a bit lost. This is particularly true for the fundamental assumption of increasing wildfire emissions, which here is stated as a matter of fact. While total burned area and even more average burned area per fire in the U.S. have increased in recent decades , it is far less clear whether burn severity has increased as well (again: Doerr and Santin 2016). And burn severity is linked to the total amount of fuel combusted which is proportional to the emissions of carbon (but not necessarily to O3, NOx etc.). For all these,burned area is a necessary but not a sufficient predictor.

The fire prediction used here by the authors is based mainly on fire weather indices. The approach is statistical, and scientifically certainly valid. However, there are other approaches that need to be mentioned and recognised. For example, the method used by the authors neglects the influence of changes in vegetation and fuel load on fire spread (please correct me if I got that wrong). But wildfires don't only need favourable fire weather to spread, they also need sufficient fuel and a continuous fuel bed. If it burns more often, there will be less fuel to burn and fire spread may be reduced. Has this negative feedback been taken into account? Has the impact of changing vegetation cover on burned area been taken into account? All these need to be better discussed.

Specific comments:

L29: This is a factual statement about the future. These should be avoided in the scientific literature.

L36: this is not 'boreal' area burned. North America does not even comprise half of the boreal zone.

L38: ambient [O3] - could this rise above critical thresholds close the active fires? The statement sounds as if it was referring to average conditions and it does not take into account the episodic nature of wildfires. This is later discussed (L350ff), but it would

be good for the reader to learn this already here.

L53: please provide more recent examples, there are plenty.

L64: I suggest dropping the topic of plant health altogether in this manuscript. Sitch et al. (2007) is about the carbon cycle and stomatal closure, and does not address the question of plant health.

L76: would drop the word "changes" here: aerosols impact the nature of the radiation, which impacts NPP. But changes in NPP do not necessarily mean changes in C uptake. This depends on changes in respiration. Needs discussion.

L95: usually, ensemble averages fear better when it comes to whether, seasonal or even decadal climate prediction. If this also applies to climate projections, however, is not something we know for sure.

L127: "The number . . . is much fewer. . ." Awkward. Better: "There are much fewer . . .".

L154: -> "A cloud mask applied to..."

L182: What is was trying to understand here is whether fuel load is constant through time. It sounds like. This is an important point that needs to be clarified and discussed through the manuscript.

L292: In addition to the observed GPP-PARdiff and GPP-PAR relationships, there should also be a sub-section on modelled GPP-PARdiff / GPP-PAR relationships. I say should, but in fact this will be crucial in order to establish the credibility of the present manuscript.

L305: This paragraph could mention that the AOD-GPP slope at CA-Gro is not significantly different form zero.

L319 "within 20%" requires continuation with "of . . .".

L417: Yes, but what about the model?

L443: I disagree. Long-term radiation changes will certainly be reflected in shade/sun adaptation of the leaves. If there is less PAR, then saturated rates of photosynthesis will decline making photosynthesis more efficient at lower rates of radiation. This is already included in the original model by Farquhar et al. (1980), which you cite here.

L467: I agree, intuitively, but I think there is no way we could quantify those uncertainties.

L516: I would really like to understand what you mean by a "missing land carbon source due to future wildfire pollution". Is the source missing now, or will it be missed in the future. And who will miss it anyway? Can you see how cloudy this statement is? But this is a good start for getting more in-depth as far as the carbon cycle is concerned (see major comments). Doesn't your model simulate the full carbon balance, including soil carbon? What happens there? Or if not, what could happen?

References:

Doerr, S. H., and Santín, C.: Global trends in wildfire and its impacts: perceptions versus realities in a changing world, Phil. Trans. R. Soc. B, 371, 20150345, 2016.

Sitch, S., Cox, P., Collins, W., and Huntingford, C.: Indirect radiative forcing of climate change through ozone effects on the land-carbon sink, Nature, 448, 791-794, 2007.

Yue, X., Mickley, L., Logan, J., Hudman, R., Martin, M. V., and Yantosca, R.: Impact of 2050 climate change on North American wildfire: consequences for ozone air quality, Atmos Chem Phys, 15, 10033-10055, 2015.

Yue, X., Mickley, L. J., Logan, J. A., and Kaplan, J. O.: Ensemble projections of wildfire activity and carbonaceous aerosol concentrations over the western United States in the mid-21st century, Atmos Environ, 77, 767-780, 2013.

---

## Referee Comment (RC2) · Anonymous Referee #2 · 11 Jul 2017

The authors discuss the hypotheses of a strong coupling between increased future biomass burning in boreal regions and feedbacks on the carbon cycle through air pollutant emissions. These feedbacks work mainly through aerosol impacts on diffuse radiation, and according to the authors less so through ozone. The aerosol feedback causes changes in atmospheric transport, leading to changing rainfall patterns and soil moisture.

While the results are overall fairly plausible, but speculative; the assumptions are not always well described and results not always sufficiently discussed.

A number of aspects of this study are particularly worrying:

- The relationship of aerosol optical thickness and NPP is based on correlations ob-

served at two stations in Canada. The correlations at these two stations are pretty weak, perhaps because there are a number of other factors that are potentially constraining NPP. The extrapolation to other boreal ecosystems is adding large additional uncertainties. This makes the study with regard to AOD highly speculative.

- The results presented in this paper are much about the feedbacks in the earth system, changes in transport etc. Yet the authors use a fairly simplified climate modeling approach in which SST is fixed, and part of the feedbacks on longer time scales are excluded. I am aware of a similar earlier paper by these authors on China, where one of the reviewers has made a similar point- and the authors asserted that these feedbacks are not dominating. But what is the evidence for that? I propose that the authors add at least one coupled ocean simulation, and resolve this issue.

- As the authors convincingly show: changes in soil moisture are dominating the carbon cycle feedbacks. However, I haven't seen at all in this publication a discussion on the accuracy of the present soil moisture simulation. Clearly a good baseline modeling of soil moisture is prerequisite for estimating these future impacts. Moreover, the authors should give a better description of what is happening with the vegetation under dryer conditions and how that in turn leads to increased fire risk and burning.

- While the authors may be right that ozone impacts is playing a minor role at high-latitudes, the discussion is very much handwaving and unconvincing. This needs to be improved.

- The uncertainties and caveats described above should be much better described in conclusions and abstract. The necessary steps in modeling and observations to corroborate the findings here should be outlined better.

Despite these shortcomings, I find the manuscript interesting and potentially important. I would therefore recommend the authors to address my major concerns and resubmit to ACP.

I have a number of more detailed comments below.

Detailed comments: l. 1 Title is not accurately describing the more limited content of the paper.

l. 29 scattering and absorption.

l. 38 The authors refer to Sitch et al (ozone flux based approach) in the text and here refer to a 40 ppb threshold- probably similar to a AOT40 type of metric. It remains unclear what has been done, and for instance which 'Sitch' (high sensivity-low sensitivity) has been used. It would be good if the authors could clarify what has been done, and show their actual stomatal ozone fluxes.

l. 43 the authors will capture only partly the feedbacks since ocean temperatures are fixed SST modelling set-up.

l. 45 How much are these direct emissions and how does it compare to the feedback effects?

L 55 see l. 45. What is found in this study and how does it relate to the air pollution change in carbon budget?

L68: more uncertain- this is a value judgement- in reality we also do not know the ozone impact well either. Perhaps what the authors want to say is that the potential impact is even larger, and can swap sign.

L81: on the other hand: to me it looks quite consistent when considering the uncertainties.

l. 82-95: part of the differences can be due to just using different climate scenarios, and are more or less comparing apples and pears. l. 95: perhaps some words why A1B- and how it maps to RCPs (I think it is RCP6.0 equivalent). In the discussion you mention that the various scenarios until 2050 it is statistically almost similar, in my experience it is the 2050s where scenarios start diverging.

[Figure]

L 113- this is a very short description. What sensitivity was included? How is consistency between the atmospheric model and land model ensured, how are fluxes calculated? Is Sitch still reflecting the newest knowledge? One can write a whole paper on what is here cryptically mentioned in one sentence.

l. 138-142 most relevant to discuss performance of MODIS retrieval over boreal areas even if it doesn't coincide with the flux sites. Summarize here what Strada found?

l. 162 would be good to provide the statistics in the supplementary and give summary here. It is really hard to understand here what is meant with 'much fewer' and how it can still be used.

l. 160-190 for clarity: future burning is assumed to be depending on fire-weather alone (regression relationship). Is there a relationship of fuel load with CO2 and fire management, if not what could be the possible uncertainties from these assumptions? Not clear here if the climate simulations would include a feedback on fires via the fire weather risk.

l. 194 please give the values. What is meant with much higher?

l. 228-232 Give a short summary on what the flux scheme is about. Summarize in a few lines what was the outcome of this benchmarking, and the consequence for this study.

l. 252 how does the RCP8.5 scenario how link to the use of the A1B scenario mentioned earlier (l 95).

l. 255 can a short description of the practical implications coming from the climate scenarios be given.

l. 258 does CO2 impact fires and fire emissions?

l. 260 Explain better the model set-up: if area is burnt, does that also change the land-cover? Would that contradict the use of prescribed landcover?

l. 273-274 2 years spin-up and 10 years seems to be a short time scale for ecosystem responses. Can the authors comment to what extent this represents full response.

l. 235-275 I would like to see a description of how the Yale model is treating regrowth after fires fires, and how the dynamics work out on time scales longer than 10 years. Can we expect an interaction between changes in age-structure and ozone and aerosol effects?

l. 287 can you show in supplementary the interpolated fields for the relevant time periods?

l. 300 would such a light saturation still be valid under changing CO2 conditions? Please comment, and what could be the impact.

l. 311 Correlation of AOD and GPP is weak to very weak. The value 3.5+/- 1.1 is just the average of the two slopes? What is the meaning of 1.1 is it one standard deviation based on two observation sets?

l. 323 Indeed patterns look everywhere reasonable except the western part. What could be the cause of this. Any indication on MODIS data quality? Or missing sources in the NASA model that can explain this? Volcanoes?

l. 326 What is compared here? 24 hr mean over June, July, August? Did the authors compare at the measurement altitude? I would recommend to focus on daytime values, as more relevant for ozone damage and usually less local conditions .The Sitch approach requires fluxes, and the methodology in this paper needs to be described as well.

l. 331-341. The increase in emissions needs to be described here. How is the contribution of wildfire emissions present determined (zero-out?). It seems that the % increase in ozone scales near-linear with the NOx (and other) emissions? But the contribution to AOD much less due to the abundance of secondary organics from BVOC emissions?

l. 344-355 the ozone damage discussion is extremely handwaving and confusing.

[Figure]

Where is the 40 ppb threshold coming from, and how does that compare to the use of the Sitch method? Only from Figure 4 I understand that indeed the Sitch high and low sensitivities have been used, but it is not discussed in the text. Anyway it seems that the model has a lot of data point above 40 ppb- but it hard to figure out how good the model performance is where the fire emissions have an impact. l. 357 at this point it is not clear what optical properties have been assigned to particles.

l. 369-381 The circulation feedbacks are an important result of the paper, but due to the approach of constraining SST will include only part of the feedbacks. I would argue that the authors should try to address in one additional simulation why they can ignore these longer timescales.

l. 402-414 We shouldn't expect a full attribution of feedbacks due to aerosol- so this is pretty convincing. However, as soil moisture is the most important feedback- I am missing here completely a discussion on how realistic soil moisture is represented in the current modeling system, and how the soil moisture feedback is leading to increased burning. At this moment a discussion of the short effects on the carbon cycle by increased burning is missing.

L 433 In Amazonia a large fraction (perhaps more than 50 %) is due to deforestation fires, and may not have a link to soil moisture. Discuss

l. 434 discrepancy or just a difference.

l. 457 Again- we need to know more about how good soil water is represented in the model- as the paper relies so much on the changes of soil moisture.

l. 439 actual->observed

l. 524 where is this number coming from.

l. 402-527 Discussion should better reflect the uncertainties of this work, and contrast them to other climatic effects on the boreal carbon cycle.

---

## Author Comment (AC1) · 25 Sep 2017

**Reviewer 1**

We are grateful to the reviewer for their time and energy in providing helpful comments and guidance that have improved the manuscript. In this document, we describe how we have addressed the reviewer's comments. Referee comments are shown in black italics and author responses are shown in blue regular text.

*The manuscript is unusual in that it considers the indirect effect of wildfires on the boreal carbon balance via emissions of atmospheric pollutants. The results are novel and the simulated effect is surprisingly large, which makes the results interesting for ACP. There is some validation of results against observations, and some evaluation of effect strength directly using site-based observations. This strengthens the paper, which otherwise relies on a very complex modelling system. I consider the subject material to be fully within scope for ACP. However, the title does not correspond to the above assessment but sets different priorities. Possibly it reflects the original idea for the manuscript on ecosystem health but the focus has changed due to the negative results regarding ozone pollution. The manuscript therefore seriously lacks focus.*

→ We appreciate the reviewer's support and helpful evaluation of this study. We agree that the original title is not appropriate for the content of the analyses. Based on the comments below, we revise the title to: "Future inhibition of ecosystem productivity by increasing wildfire pollution over boreal North America" so as to better reflect the main focus of the study.

*Major comments:*

*I can see at least the following scientific questions either being addressed, or requiring attention:*

*1) Do wildfires affect ecosystem health in boreal environments beyond its direct impact through higher ozone concentrations, i.e. far away from the fire or long after the fire has ended?*
*2) How can the effect be quantified, i.e. is NPP a valid proxy?*
*3) How do the direct and indirect effects compare?*
*4) Will the strength of this effect change in the future?*
*5) How do wildfires affect the carbon balance of boreal environments indirectly through atmospheric pollution away from the burned area?*
*6) How does this effect compare to the direct effect on the carbon cycle in the burned area?*
*7) How will this change in the future?*
*8) Are the results representative of all boreal regions?*

*Questions 1-4 correspond to the title, but Questions 5-8 to the actual focus of the paper (but still not all of them are being answered).*

*In order to become publishable, either the title needs to be changed to reflect the true focus of the paper, or the focus of the paper needs to be changed and much more detail on ecosystem health effects need to be included. The latter is probably beyond scope, so the best way forward must be the former. In that case, however, more depth is required regarding the carbon cycle, as NPP is only one of many components, and all of Questions 5-8 need to be answered. If the impact on the carbon cycle were to be the focus, then the title would have to be adapted and the manuscript would have to include more discussion that puts the results into the perspective of the regional and global carbon cycle. Some of it is there, but not enough to give the reader a sufficiently good feel for how important this really is. So if the focus is to be on the carbon cycle, more results need to be included or a more detailed and in-depth discussion is needed. Or, what is also possible, restrict the paper to impacts on NPP alone. After all, you also include GPP, and that is already a step that involves changes in plant respiration, which also need to be projected. What happens here?*

→ Questions 1-4 are related to ecosystem health while the main focus of this study is the responses of ecosystem primary productivity (including both GPP and NPP) to the combined effects of fire pollutants. As a result, the words 'ecosystem health' used in the original title are not appropriate. We have changed the title to "Future inhibition of ecosystem productivity by increasing wildfire pollution over boreal North America" to reflect the main objective of this study.

For questions 5-8, we answered and/or discussed them substantially in the paper. Questions 5 and 7 are the main focus of the study and their answers have been shown in Figure 12, with model evaluations in Figures 2-5 as the solid basis. For question 6, we discussed it in the last section: "Fire pollution aerosol increases boreal NPP by 72 Tg C yr$^{-1}$ in the present day, comparable to the direct carbon loss of 68 Tg C yr$^{-1}$ from wildfire $CO_2$ emissions (product of biomass burned and $CO_2$ emission factors). By midcentury, increasing fire emissions instead cause a NPP reduction of 118 Tg C yr$^{-1}$ due to the amplified drought. Although NPP is not a direct indicator of the land carbon sink, reduction of NPP is always accompanied with the decline of net ecosystem exchange (NEE) and the enhanced carbon loss. In combination with the enhanced carbon emission of 130 Tg C yr$^{-1}$, future boreal wildfire presents an increasing threat to the regional carbon balance and global warming mitigation." (Lines 673-681)

For question 8, we discussed it as follows: "Our analyses of fire pollution effects on boreal North American productivity may not be representative for other boreal ecosystems and/or on the global scale. There is substantial variability in plant species, topography, and climatology across different boreal regions. Such differences indicate distinct GPP sensitivities as well as fire characteristics. At lower latitudes, where anthropogenic pollution emissions are more abundant, ambient ozone concentrations may have exceeded damaging thresholds for most plant species. In those regions, additional ozone from a fire plume may cause more profound impacts on photosynthesis than our estimate for boreal North America. For example, Amazonian fire is predicted to reduce forest NPP by 230 Tg C yr$^{-1}$ through the generation of surface ozone (Pacifico et al., 2015). Meanwhile, solar radiation is more abundant at lower latitudes, indicating more

efficient increases in photosynthesis through aerosol DFE because the sunlit leaves receive saturated direct light in those regions. As shown in Beer et al. (2010), partial correlations between GPP and solar radiation are positive in boreal regions but negative over the subtropics/tropics, suggesting that light extinction by fire aerosols has contrasting impacts on plant photosynthesis in the high versus low latitudes. Further simulations and analyses are required to understand the net impacts of ozone and aerosols from biomass burning on the global carbon cycle." (Lines 685-701)

In the revised paper, we show the changes in plant respiration in Figure S2. We found that: "Such changes in NPP are a consequence of changes in GPP and autotrophic respiration (Fig. S2). Variations in plant respiration resemble those of GPP, because higher photosynthesis leads to faster leaf/tissue development, resulting larger maintenance and growth respiration." (Lines 496-499)

*Finally, the result must be backed up more by measurements. The main effect is surprising, but it will be crucial that there is a thorough evaluation of how the model simulates the impact of changes in diffuse and direct light on GPP, as opposed to the measurements.*

→ In the revised paper, we performed additional validations by conducting two new simulations at sites CA-Gro and CA-Qfo. The simulated GPP responses to diffuse and direct PAR are consistent with observations as shown in Figure 5, suggesting that the model can reasonably capture changes in GPP due to aerosol-induced perturbations in radiation.

"The model also reproduces observed light responses of GPP to diffuse radiation in boreal regions. With the site-level simulations, we evaluate the modeled GPP-$PAR_{dif}$ relationships at the hourly (instead of half-hourly) time step during summer. For 1342 pairs of GPP and $PAR_{dif}$ at the site CA-Gro, the observed correlation coefficient is 0.42 and regression slope is 0.011, while the results for the simulation are 0.60 and 0.014, respectively. At the site CA-Qfo, the observations yield a correlation coefficient of 0.46 and regression slope of 0.007 for 1777 pairs of GPP and $PAR_{dif}$. The simulated correlation is 0.61 and the regression is 0.011 at the same site. The GPP sensitivity to $PAR_{dif}$ in the model is slightly higher than that of the available observations, likely because the latter are affected by additional non-meteorological abiotic factors. To remove the influences of compound factors other than radiation, we follow the approach of Mercado et al. (2009) to discriminate GPP responses to 'diffuse' and 'direct' components of PAR at the two sites (Fig. 5). The model successfully reproduces the observed GPP-to-PAR sensitivities. Increase in PAR boosts GPP, but the efficiency is much higher for diffuse light than that for direct light, suggesting that increase of diffuse radiation is a benefit for plant growth." (Lines 415-430)

*The third possibility would be to simply focus on the effect of atmospheric pollution from wildfires on GPP (not NPP) in boreal North America (and change the title accordingly).*

→ Yes, we have changed the title to "Future inhibition of ecosystem productivity by increasing wildfire pollution over boreal North America" to reflect the main objective of this study.

*Another major comment: the different chains of events discussed here are enormously complicated and the effect is very indirect. I suggest the authors show this in a suitable graphic. We have changes in climate affecting fire weather, but also affecting vegetation composition and fuel load. In addition we have changes in land use, in particular forestry and fire management (See Fig. 3 in Doerr and Satin showing for the U.S. increasing burned area, fewer fires, and an enormous rise in fire suppression costs). Both impact burned area and fire emissions. But then we also have atmospheric circulation patterns which are influenced by all sorts of things, among them greenhouse gas concentrations and aerosol load, some of it from boreal forest fires. And all of these together influence boreal forest NPP which in turn impacts the regional and global carbon cycle. Given this enormously complex web of causes and effects, I am not sure what we really learn here. It us up to the authors to clarify and give us a clear picture of what this paper is really about. Do that, I suggest considering the main questions and sub-questions as above, and then re-structuring the paper in order to answer them all in a systematic way. Much of it is there, but the information is too scattered.*

→We agree that multiple factors, including climate change, land use change, and human activities (forest management) will affect both wildfire and ecosystem productivity. As the reviewer commented, these processes interplay with each other, leading to large uncertainties in the estimate. For the current study, we clarify that we limit our focuses to the processes shown in Figure 1. For other indirect processes, we either use fixed values for present day and midcentury (e.g., fuel load and vegetation cover) or ignore the related impacts due to the large uncertainties (e.g., forest management). In the discussion section, we explained why we used fixed fuel load and vegetation cover (Lines 624-640, or see the following response).

In the Introduction section, we present a new Figure 1 to clarify the main processes we examined in this study: "The major chain we investigate includes i) generation of aerosols and surface ozone from wildfire emissions and ii) impact of fire-emitted aerosols and ozone on plant photosynthesis through physical and biogeochemical processes." (Lines 116-118)

*Increases in boreal wildfire activity: this manuscript builds heavily on Yue et al. (2015), which in turn builds heavily on Yue et al. (2013). This compartmentalisation of research is necessary given the said complexity of the subject. However, the foundations and basic assumptions on which the story rests here get a bit lost. This is particularly true for the fundamental assumption of increasing wildfire emissions, which here is stated as a matter of fact. While total burned area and even more average burned area per fire in the U.S. have increased in recent decades, it is far less clear whether burn severity has*

*increased as well (again: Doerr and Santin 2016). And burn severity is linked to the total amount of fuel combusted which is proportional to the emissions of carbon (but not necessarily to O3, NOx etc.). For all these, burned area is a necessary but not a sufficient predictor.*

→ Yes, we built the fire projections on the previous studies of Yue et al. (2013) and Yue et al. (2015). The decision is justified because of the complexity of this interdisciplinary research and because those previous published studies underwent rigorous uncertainty analysis. However, to avoid the confusion mentioned here, and to make this study complete and independent from earlier work, we explained more details about fire prediction and the foundations of our assumptions in the revised paper.

For this study, we apply constant fuel load for both present day and midcentury, but we consider impacts of climate change on fuel consumption by implementing responses of fuel moisture. As we discussed in section 4.2, changes in area burned likely dominate the projected changes in fire emissions:
"We apply constant land cover and fuel load for both present day and midcentury, but we estimate an increase in fuel consumption due to changes in fuel moisture. Future projection of boreal fuel load is highly uncertain because of multiple contrasting influences. For example, using a dynamic global vegetation model (DGVM) and an ensemble of climate change projections, Heyder et al. (2011) predicted a large-scale dieback in boreal-temperate forests due to increased heat and drought stress in the coming decades. On the contrary, projections using multiple DGVMs show a widespread increase in boreal vegetation carbon under the global warming scenario with $CO_2$ fertilization of photosynthesis (Friend et al., 2014). In addition, compound factors such as greenhouse gas mitigation (Kim et al., 2017), pine beetle outbreak (Kurz et al., 2008), and fire management (Doerr and Santin, 2016) may exert varied impacts on future vegetation and fuel load. Although we apply constant fuel load, we consider changes of fuel moisture because warmer climate states tend to dry fuel and increase fuel consumption (Flannigan et al., 2016). With constant fuel load but climate-driven fuel moisture, we calculate a 9% increase in boreal fuel consumption by the midcentury (Yue et al., 2015). Although such increment is higher than the prediction of 2-5% by Amiro et al. (2009) for a doubled-$CO_2$ climate, the consumption-induced uncertainty for fire emission is likely limited because changes in area burned are much more profound." (Lines 624-640)

Fire emission is largely dependent on area burned, not only because the amount of biomass burned is in ratio to area burned, but also because the larger area burned usually causes higher severity (Turetsky et al., 2011). From this aspect, larger area burned may have both positive (higher severity) and negative (fewer fuel left) impacts on emissions. For this study, we select area burned as the main metric to reduce the possible uncertainties in the estimate of fire emissions.

*The fire prediction used here by the authors is based mainly on fire weather indices. The approach is statistical, and scientifically certainly valid. However, there are other*

*approaches that need to be mentioned and recognised. For example, the method used by the authors neglects the influence of changes in vegetation and fuel load on fire spread (please correct me if I got that wrong). But wildfires don't only need favourable fire weather to spread, they also need sufficient fuel and a continuous fuel bed. If it burns more often, there will be less fuel to burn and fire spread may be reduced. Has this negative feedback been taken into account? Has the impact of changing vegetation cover on burned area been taken into account? All these need to be better discussed.*

→ For this study, we predict area burned on the ecoregion basis. In each ecoregion, similar impacts of topography, human activity, and vegetation (fuel types and load) on the spread of wildfires are expected. This approach facilitates the comparisons of area burned in the present day and the future climate for regions with varied landscape features. Analyses of multiple observations have shown that weather parameters play the dominant role in regulating fire activity in boreal ecoregions (Gillett et al., 2004; Flannigan et al., 2005; Fauria and Johnson, 2006; Girardin and Wotton, 2009; Meyn et al., 2010), supporting the concept of fire prediction using weather factors/indexes.

We agree that other non-climatic factors influence wildfire ignition and spread efficiency. For example, fuel changes will alter the possibility of fire occurrence. These interactive processes are hardly included in a fire-weather model but could be considered in dynamic global vegetation models (DGVM). However, large uncertainties and complex feedbacks will diminish the credibility of fire predictions from DGVMs. For example, using different DGVMs, order of magnitude differences in the area burned changes are predicted over the U.S. (Bachelet et al., 2003; Rogers et al., 2011). In addition, for many DGVMs, the present-day area burned is not validated against observations. Furthermore, coupling an interactive fire scheme to the dynamic carbon cycle as a disturbance is a relatively new emerging research area. Meanwhile, the regressions used by Yue et al. (2015) explain 34-75% of variances of boreal area burned during 1980-2009.

In the revised paper, we explained how to consider the impact of fuel availability on fire spread as follows:
"We derive 1°×1° gridded area burned based on the prediction for each ecoregion following the approach by Yue et al. (2015). Temporally, the annual area burned estimated with regressions is first converted to monthly area burned using the mean seasonality for each boreal ecoregion during 1980-2009. Spatially, large fires tend to burn in ecosystems where historical fires are frequent because of favorable conditions (Keane et al., 2008). In each 1°×1° grid square, we calculate the frequency of fires larger than 1000 ha during 1980-2009; these fires account for about 85% of total area burned in boreal North America. We arbitrarily attribute 85% of area burned within each ecoregion to a number of fires with fixed size of 1000 ha. We then allocate these large fires among the 1°×1° grid cells based on the observed spatial probability of large fires. For example, if one grid box (named grid 'A') bears 1% of large fires (>1000 ha) within an ecoregion at present day, the same grid will bear the same possibility for large fires in the future. On the other hand, fuel availability limits reburning and fire spread during the forest return interval, suggesting that local burning will decrease the possibility of fires in the same location. To consider such impact, we scale the observed probabilities by the fraction

remaining unburned in each grid box, and then use this modified probability distribution to allocate large fires for the remaining months. For example, if present-day fires have consumed 20% of the total area within the grid 'A', then the possibility of large fire will be 0.8% (1%×0.8, instead of 1%) for this grid. Finally, we disaggregate the remaining 15% of area burned into fires 10 ha in size, and randomly distribute these fires across all grid boxes in the ecoregion. With this method, we derive the gridded area burned for boreal North America by eliminating reburning issues. Sensitivity tests show that specifying different area burned to the large fires (100 or 10 000 ha rather than 1000 ha) yields < 1 % changes in predicted biomass burned, suggesting that this approach is not sensitive to the presumed fire size in the allocation procedure." (Lines 204-227)

**Specific comments:**

*L29: This is a factual statement about the future. These should be avoided in the scientific literature.*

➔ We changed the statement to: "Wildfire area burned is projected to increase significantly in boreal North America by the midcentury" (Lines 32 – 33)

*L36: this is not 'boreal' area burned. North America does not even comprise half of the boreal zone.*

➔ We changed the sentence to: "area burned is predicted to increase by 66% in boreal North America" (Lines 39-40).

*L38: ambient [O3] - could this rise above critical thresholds close the active fires? The statement sounds as if it was referring to average conditions and it does not take into account the episodic nature of wildfires. This is later discussed (L350ff), but it would be good for the reader to learn this already here.*

➔ We clarified as follows: "Fire $O_3$ causes negligible impacts on NPP because ambient $O_3$ concentration (with fire contributions) is below the damage threshold of 40 ppbv for 90% summer days." (Lines 40-42)

*L53: please provide more recent examples, there are plenty.*

➔ We added two recent examples (Groot et al., 2013; Wang et al., 2015) as suggested.
Groot, W. J. d., D.Flannigan, M., and S.Cantin, A.: Climate change impacts on future boreal fire regimes, Forest Ecology and Management, 294, doi:10.1016/j.foreco.2012.09.027, 2013.
Wang, X., Thompson, D. K., Marshall, G. A., Tymstra, C., Carr, R., and Flannigan, M. D.: Increasing frequency of extreme fire weather in Canada with climate change, Climatic Change, 130, 573-586, doi:10.1007/s10584-015-1375-5, 2015.

*L64: I suggest dropping the topic of plant health altogether in this manuscript. Sitch et al. (2007) is about the carbon cycle and stomatal closure, and does not address the question of plant health.*

→ We revised this sentence as follows: "Surface O3 causes damages to photosynthesis through stomatal uptake (Sitch et al., 2007)" The words "plant health" have been removed throughout the paper.

*L76: would drop the word "changes" here: aerosols impact the nature of the radiation, which impacts NPP. But changes in NPP do not necessarily mean changes in C uptake. This depends on changes in respiration. Needs discussion.*

→ We revised this sentence as follows: "Furthermore, the aerosol radiative effects indirectly influence ecosystem productivity through concomitant meteorological perturbations that are only beginning to be examined" (Lines 81-83)

For this study, we show the responses in respiration in Fig. S2.

*L95: usually, ensemble averages fear better when it comes to whether, seasonal or even decadal climate prediction. If this also applies to climate projections, however, is not something we know for sure.*

→ We revised the sentence as follows:
"The multi-model ensemble approach has shown superior predictability over single models in historical climate simulations (Flato et al., 2013) and near-term climate predictions (Kirtman et al., 2014), and has been used as a standard technique to assess changes of climate variables in the long-term projections (Collins et al., 2013)." (Lines 99-103)

*L127: "The number . . . is much fewer. . ." Awkward. Better: "There are much fewer . . .".*

→ Revised as suggested.

*L154: -> "A cloud mask applied to..."*

→ Revised as suggested.

*L182: What is was trying to understand here is whether fuel load is constant through time. It sounds like. This is an important point that needs to be clarified and discussed through the manuscript.*

→ Yes, we use constant fuel load for this study because of large uncertainties exist for fuel projection. Instead, we consider changes in burning severity due to perturbations in fuel moisture.

In section "2.2 Wildfire emissions", we explained as follows:
"As in Amiro et al. (2009) and Yue et al. (2015), we apply constant fuel load for both present day and midcentury because opposite and uncertain factors influence future projections (Kurz et al., 2008; Heyder et al., 2011; Friend et al., 2014; Kim et al., 2017). Instead, we consider changes in burning severity due to perturbations in fuel moisture as indicated by CFWI indexes (Yue et al., 2015). On average, we estimate a 9% increase in fuel consumption over boreal North America by the midcentury, because higher temperature and lower precipitation result in a future with drier fuel load (Flannigan et al., 2016)." (Lines 237-244)

In section "4.2 Limitations and uncertainties" we discussed as follows:
"We apply constant land cover and fuel load for both present day and midcentury, but we estimate an increase in fuel consumption due to changes in fuel moisture. Future projection of boreal fuel load is highly uncertain because of multiple contrasting influences. For example, using a dynamic global vegetation model (DGVM) and an ensemble of climate change projections, Heyder et al. (2011) predicted a large-scale dieback in boreal-temperate forests due to increased heat and drought stress in the coming decades. On the contrary, projections using multiple DGVMs show a widespread increase in boreal vegetation carbon under the global warming scenario with $CO_2$ fertilization of photosynthesis (Friend et al., 2014). In addition, compound factors such as greenhouse gas mitigation (Kim et al., 2017), pine beetle outbreak (Kurz et al., 2008), and fire management (Doerr and Santin, 2016) may exert varied impacts on future vegetation and fuel load. Although we apply constant fuel load, we consider changes of fuel moisture because warmer climate states tend to dry fuel and increase fuel consumption (Flannigan et al., 2016). With constant fuel load but climate-driven fuel moisture, we calculate a 9% increase in boreal fuel consumption by the midcentury (Yue et al., 2015). Although such increment is higher than the prediction of 2-5% by Amiro et al. (2009) for a doubled-$CO_2$ climate, the consumption-induced uncertainty for fire emission is likely limited because changes in area burned are much more profound." (Lines 624-640)

*L292: In addition to the observed GPP-PARdiff and GPP-PAR relationships, there should also be a sub-section on modelled GPP-PARdiff / GPP-PAR relationships. I say should, but in fact this will be crucial in order to establish the credibility of the present manuscript.*

→ In section 3.2, we added additional model validation for GPP-PARdif relationships:
"The model also reproduces observed light responses of GPP to diffuse radiation in boreal regions. With the site-level simulations, we evaluate the modeled GPP-PAR$_{dif}$ relationships at the hourly (instead of half-hourly) time step during summer. For 1342 pairs of GPP and PAR$_{dif}$ at the site CA-Gro, the observed correlation coefficient is 0.42 and regression slope is 0.011, while the results for the simulation are 0.60 and 0.014, respectively. At the site CA-Qfo, the observations yield a correlation coefficient of 0.46 and regression slope of 0.007 for 1777 pairs of GPP and PAR$_{dif}$. The simulated

correlation is 0.61 and the regression is 0.011 at the same site. The GPP sensitivity to PAR$_{dif}$ in the model is slightly higher than that of the available observations, likely because the latter are affected by additional non-meteorological abiotic factors. To remove the influences of compound factors other than radiation, we follow the approach of Mercado et al. (2009) to discriminate GPP responses to 'diffuse' and 'direct' components of PAR at the two sites (Fig. 5). The model successfully reproduces the observed GPP-to-PAR sensitivities. Increase in PAR boosts GPP, but the efficiency is much higher for diffuse light than that for direct light, suggesting that increase of diffuse radiation is a benefit for plant growth." (Lines 415-430)

*L305: This paragraph could mention that the AOD-GPP slope at CA-Gro is not significantly different form zero.*

→ We clarified as follows:
"However, the slope of regression between GPP and AOD is lower (and not significant) at CA-Gro compared with that at CA-Qfo" (Lines 384-385)

*L319 "within 20%" requires continuation with "of . . .".*

→ We revised the sentence as follows:
"Simulated GPP reasonably captures the spatial distribution with a high correlation coefficient of 0.77 (*p* << 0.01) and relatively small biases within 20% of the data product." (Lines 395-397)

*L417: Yes, but what about the model?*

→ We have validated the modeled GPP-PARdif relationships in the revised paper (see section 3.2).

*L443: I disagree. Long-term radiation changes will certainly be reflected in shade/sun adaptation of the leaves. If there is less PAR, then saturated rates of photosynthesis will decline making photosynthesis more efficient at lower rates of radiation. This is already included in the original model by Farquhar et al. (1980), which you cite here.*

→ Photosynthesis might be more efficient if PAR is reduced on the long-term period. However, such acclimation of photosynthesis is not unlimited. The validation in Figure 5 shows that the model (using Farquhar-Ball-Berry scheme) can reasonably capture GPP responses if both direct and diffuse radiation is reasonable. At the high latitudes, solar radiation is less abundant compared with that at lower latitudes. As a result, sunlit leaves at boreal regions are more sensitive to the reduction of direct light, offsetting the benefit of increased diffuse light. Observations also support this conclusion. "As shown in Beer et al. (2010), partial correlations between GPP and solar radiation are positive in boreal regions but negative over the subtropics/tropics, suggesting that light extinction by fire

aerosols has contrasting impacts on plant photosynthesis in the high versus low latitudes." (Lines 696-699).

*L467: I agree, intuitively, but I think there is no way we could quantify those uncertainties.*

→ As in the Introduction section, we include citations to support the statement:
"Such an approach may help reduce model uncertainties in climatic responses to $CO_2$ changes (Collins et al., 2013; Kirtman et al., 2014), …" (Lines 602-604)

*L516: I would really like to understand what you mean by a "missing land carbon source due to future wildfire pollution". Is the source missing now, or will it be missed in the future. And who will miss it anyway? Can you see how cloudy this statement is? But this is a good start for getting more in-depth as far as the carbon cycle is concerned (see major comments). Doesn't your model simulate the full carbon balance, including soil carbon? What happens there? Or if not, what could happen?*

→ We agree that the expression "missing land carbon" was somewhat vague in the original manuscript version. We emphasize that fire pollution dampens land carbon assimilation in the 'future', instead of 'present day'. The climate model ModelE2-YIBs includes full carbon cycle for land ecosystem, but the current version does not include dynamic atmospheric $CO_2$ or dynamic ocean $CO_2$ cycle. The soil respiration takes thousands of years to reach equilibrium in the model, evolves on much longer timescales than air pollution chemistry (centuries/millennia versus years/decades), and requires transient versus time-slice simulations. Therefore, we made a decision to focus on ecosystem productivity, rather than the longer-term land carbon storage, as our metric of impact. In the discussion, we clarify that NPP is different from NEE but can be used as an indicator for the ecosystem carbon uptake: "Although NPP is not a direct indicator of the land carbon sink, reduction of NPP is always accompanied with the decline of net ecosystem exchange (NEE) and the enhanced carbon loss." (Lines 677-679)

[revised manuscript text omitted]

---

## Author Comment (AC2) · 25 Sep 2017

**Reviewer 2**

We are grateful to the reviewer for their time and energy in providing helpful comments and guidance that have improved the manuscript. In this document, we describe how we have addressed the reviewer's comments. Referee comments are shown in black italics and author responses are shown in blue regular text.

*The authors discuss the hypotheses of a strong coupling between increased future biomass burning in boreal regions and feedbacks on the carbon cycle through air pollutant emissions. These feedbacks work mainly through aerosol impacts on diffuse radiation, and according to the authors less so through ozone. The aerosol feedback causes changes in atmospheric transport, leading to changing rainfall patterns and soil moisture.*

*While the results are overall fairly plausible, but speculative; the assumptions are not always well described and results not always sufficiently discussed.*

*A number of aspects of this study are particularly worrying:*

*- The relationship of aerosol optical thickness and NPP is based on correlations observed at two stations in Canada. The correlations at these two stations are pretty weak, perhaps because there are a number of other factors that are potentially constraining NPP. The extrapolation to other boreal ecosystems is adding large additional uncertainties. This makes the study with regard to AOD highly speculative.*

→ In the revised paper, we performed two new simulations at sites CA-Gro and CA-Qfo. The simulated GPP responses to diffuse and direct PAR are consistent with observations as shown in Figure 5, suggesting that the model can reasonably capture changes in GPP due to aerosol-induced perturbations in radiation.

"The model also reproduces observed light responses of GPP to diffuse radiation in boreal regions. With the site-level simulations, we evaluate the modeled GPP-PAR$_{dif}$ relationships at the hourly (instead of half-hourly) time step during summer. For 1342 pairs of GPP and PAR$_{dif}$ at the site CA-Gro, the observed correlation coefficient is 0.42 and regression slope is 0.011, while the results for the simulation are 0.60 and 0.014, respectively. At the site CA-Qfo, the observations yield a correlation coefficient of 0.46 and regression slope of 0.007 for 1777 pairs of GPP and PAR$_{dif}$. The simulated correlation is 0.61 and the regression is 0.011 at the same site. The GPP sensitivity to PAR$_{dif}$ in the model is slightly higher than that of the available observations, likely because the latter are affected by additional non-meteorological abiotic factors. To remove the influences of compound factors other than radiation, we follow the approach of Mercado et al. (2009) to discriminate GPP responses to 'diffuse' and 'direct' components of PAR at the two sites (Fig. 5). The model successfully reproduces the observed GPP-to-PAR sensitivities. Increase in PAR boosts GPP, but the efficiency is much higher for diffuse light than that for direct light, suggesting that increase of diffuse radiation is a benefit for plant growth." (Lines 415-430)

We extrapolate the AOD-GPP relationships at two sites as representative of North American boreal ecosystems because of the limitation in data availability. The weak correlations between AOD and GPP are observational results. Through comprehensive validation with all available observational data for carbon fluxes, air pollution concentrations, and GPP sensitivities to ozone and diffuse radiation (section 3.2), we assert that our results have been constrained to measurements/observations to the maximum extent possible. Let us reflect that global coupled Earth system models exist exactly to probe the types of underlying process interactions and feedbacks in this study, where it is fundamentally impossible to "see" the effect in observations alone that by nature integrate all processes simultaneously.

*- The results presented in this paper are much about the feedbacks in the earth system, changes in transport etc. Yet the authors use a fairly simplified climate modeling approach in which SST is fixed, and part of the feedbacks on longer time scales are excluded. I am aware of a similar earlier paper by these authors on China, where one of the reviewers has made a similar point- and the authors asserted that these feedbacks are not dominating. But what is the evidence for that? I propose that the authors add at least one coupled ocean simulation, and resolve this issue.*

The referee misunderstands some aspects of the Earth system model experimental design. The "issue" is not going to be resolved by adding "at least one coupled ocean simulation."

→ Firstly, it is a common and valid approach to investigate regional aerosol-climate feedbacks without ocean responses. For example, Cook et al. (2009) found that dust-climate-vegetation feedback promotes drought in U.S., with a climate model driven by prescribed SSTs. Similarly, Liu (2005) found fire aerosols enhance regional drought using a regional climate model, which even ignores the feedback between local climate and large-scale circulation. Regional climate model frameworks such as WRF-Chem are regularly applied to understand effects of aerosol pollution on weather patterns under the assumption of fixed SSTs. Ocean feedbacks are important but slow (century/millennial), while aerosol effects over land are usually fast (annual/decadal). Applying fixed SSTs, which is the fundamental basis of the Effective Radiative Forcing metric defined in IPCC AR5, allows us to explore the complex system step by step.

Secondly, running with a fully coupled dynamic ocean would require a several-thousands-of-years preindustrial spin-up, followed by several ensemble-member transient preindustrial to present-day runs. We do not have access to the computational resources required for such dynamic ocean simulations that are generally in the remit of the international climate modelling centers. For example, GISS performs these simulations with ModelE2 for the CMIP, but no simulations are available with our coupled vegetation model YIBs. Furthermore, inclusion of dynamical ocean feedbacks might introduce additional uncertainties to the system, making it difficult to identify the direct impact of aerosols.

Thirdly, slab ocean simulations are not viable either because we do not have projections of mixed layer depth by 2050s, which might change substantially, but very uncertain for different CGCMs (Yeh et al., 2009). Therefore, it is not possible to obtain the associated future atmosphere-ocean heat fluxes for our time-slice simulations. The future 2050 time-slice projections in our work do apply future SSTs and sea ice boundary conditions.

Finally, the reviewer connected this question to our recent publication focused on China. Actually, the referee of that paper had some concerns on the dynamical large-scale signals between regional and global scales, though he considered the use of fixed SSTs might introduce exaggerated responses over land due to the artificial land-ocean thermal contrast. Our responses to that paper did not deny such deficit: "Diagnosing long range dynamical mechanisms is out of scope of this study, …, this specific study will not gain from an explicit description of the multi-scale dynamical mechanisms that drive the regional meteorological changes". In another recent study, that was focused at the global-scale, however, we have identified the separate and combined roles of fast aerosol feedbacks associated with the land and slow aerosol feedbacks associated with the ocean: "Unger N, Yue X, Harper KL. (2017) Aerosol climate change effects on land ecosystem services, *Faraday Discuss*, 200, 121-142, DOI:10.1039/C7FD00033B."

*- As the authors convincingly show: changes in soil moisture are dominating the carbon cycle feedbacks. However, I haven't seen at all in this publication a discussion on the accuracy of the present soil moisture simulation. Clearly a good baseline modeling of soil moisture is prerequisite for estimating these future impacts. Moreover, the authors should give a better description of what is happening with the vegetation under dryer conditions and how that in turn leads to increased fire risk and burning.*

→ Global observations of soil moisture are not available. In the revised paper, we compare soil moisture with two different datasets in Figure S1. The comparisons show that the ModelE2-YIBs model generally reproduces the reasonable spatial pattern with low biases. "For >3300 land grids in the summer, the spatial correlation coefficient is $R = 0.25$ between ModelE2-YIBs and CLM, and $R = 0.34$ between CLM and ERA-Interim. The global area-weighted soil moisture is 0.22 $mm^3$ $mm^{-3}$ for ModelE2-YIBs, 0.26 $mm^3$ $mm^{-3}$ for CLM, and 0.23 $mm^3$ $mm^{-3}$ for ERA-Interim. Statistics for winter are very similar to the summer results."

*- While the authors may be right that ozone impacts is playing a minor role at high-latitudes, the discussion is very much handwaving and unconvincing. This needs to be improved.*

In the revised text, we clarified that:
"The impacts of the boreal fire $O_3$ on forest photosynthesis are predicted using the flux-based damage algorithm proposed by Sitch et al. (2007), which has been fully evaluated

against available $O_3$ damage sensitivity measurements globally and over North America (Yue and Unger, 2014; Yue et al., 2016; Yue et al., 2017)" (Lines 125-128)

We explained how Sitch's scheme works:
"For this scheme, $O_3$ damaging level is dependent on excess $O_3$ stomatal flux within leaves, which is a function of ambient $O_3$ concentration, boundary layer resistance, and stomatal resistance. Reduction of photosynthesis is calculated on the basis of plant functional types (PFTs), each of which bears a range of low-to-high sensitivities to $O_3$ uptake." (Lines 290-294).

We summarized the evaluation of Sitch's scheme:
"With the Sitch et al. (2007) scheme, the YIBs model simulates reasonable GPP responses to $[O_3]$ in North America (Yue and Unger, 2014; Yue et al., 2016). Generally, damage to GPP increases with the enhancement of ambient $[O_3]$, but with varied sensitivities for different plant species (see Fig. 6 of Yue and Unger (2014)). In response to the same level of $[O_3]$, predicted $O_3$ damages are higher for deciduous trees than that for needleleaf trees, consistent with observations from meta-analyses (Wittig et al., 2007)." (Lines 410-415)

In the following responses, we showed the validation of Sitch et al. (2007) scheme globally and regionally (Figures R1 and R2), which we did not present in the paper because those plots have been published in our previous work.

Finally, we show $O_3$ stomatal flux in a new Figure 8, which shows that $O_3$ uptake is limited in boreal North America.

All these results support our conclusion that $O_3$ vegetation damage, no matter including fire emissions or not, is trivial over boreal North America.

*- The uncertainties and caveats described above should be much better described in conclusions and abstract. The necessary steps in modeling and observations to corroborate the findings here should be outlined better.*

→ We extend the discussion about the uncertainties and caveats of the research: "In this study, we examine the interactions among climate change, fire activity, air pollution, and ecosystem productivity. To reduce the complexity of the interactions, we focus on the most likely dominant feedback and thus main chain of events: "climate → fire → pollution → biosphere' (Fig. 1). However, our choice of feedback analysis does not mean that the interplay of other processes is unimportant. For example, climate-induced changes in vegetation cover/types can influence fire activity by alteration of fuel load, and air pollution by BVOC emissions (climate → biosphere → fire/pollution). In addition, other feedbacks may amplify ecosystem responses but are not considered. For example, the drought caused by fire aerosols in the midcentury (Fig. 11) may help increase fire activity (fire → pollution → climate → fire). Furthermore, we apply fixed SSTs in the climate simulations because reliable ocean heat fluxes for the future world were not

available. Many previous studies have investigated regional aerosol-climate feedbacks without ocean responses. For example, Cook et al. (2009) found that dust-climate-vegetation feedback promotes drought in U.S., with a climate model driven by prescribed SSTs. Similarly, Liu (2005) found fire aerosols enhance regional drought using a regional climate model, which even ignores the feedback between local climate and large-scale circulation. While we do concede that our experimental design is not a complete assessment of all known processes and feedbacks, within these limitations, this study for the first time quantifies the indirect impacts of wildfire on long-range ecosystem productivity under climate change." (Lines 581-599).

Additional model validations (Figure 5) have been performed to corroborate the main findings of this research.

*Despite these shortcomings, I find the manuscript interesting and potentially important. I would therefore recommend the authors to address my major concerns and resubmit to ACP.*

*I have a number of more detailed comments below.*

*Detailed comments:*

*l. 1 Title is not accurately describing the more limited content of the paper.*

→ The title has been changed to "Future inhibition of ecosystem productivity by increasing wildfire pollution over boreal North America" to reflect the main focus of the study.

*l. 29 scattering and absorption.*

→ Revised as suggested.

*l. 38 The authors refer to Sitch et al (ozone flux based approach) in the text and here refer to a 40 ppb threshold- probably similar to a AOT40 type of metric. It remains unclear what has been done, and for instance which 'Sitch' (high sensitivity-low sensitivity) has been used. It would be good if the authors could clarify what has been done, and show their actual stomatal ozone fluxes.*

→ Yes, we used Sitch et al. (2007) scheme for this study. In our previous work, we have validated Sitch's scheme against available observations. Figure R1 is adopted from Yue and Unger (2014), which shows percentage changes in GPP of different PFTs over North America in response to varied levels of [O3]. Square symbols are from measurements. As Figure R1 shows, evergreen needleleaf forest (ENF) and shrubland (SHR), which are dominant PFTs over boreal North America, have low sensitivity to $O_3$ damages with a damaging threshold of 40 ppbv. In the paper, we explained more details about $O_3$ thresholds. We also show the stomatal ozone fluxes in a new figure 8 as suggested.

"Surface O3, including both fire and non-fire emissions (Table 2), causes limited (1-2%) damages to summer GPP in boreal North America (Fig. 7)." (Lines 452-453).

"Over boreal North America, dominant PFTs are ENF (accounting for 44% of total vegetation cover) and tundra (treated as shrubland, accounting for 41% of total vegetation cover). Both species have shown relatively high $O_3$ tolerance with a damaging threshold of 40 ppbv as calculated with Sitch's scheme (Yue and Unger, 2014). For boreal regions, the mean $[O_3]$ of 28 ppbv (Fig. 4a) is much lower than this damaging threshold, explaining why the excess $O_3$ stomatal flux (the flux causing damages) is low there (Fig. 8)." (Lines 457-462).

[Figure]

**Figure R1.** Changes in GPP for all and individual PFTs in the presence of different levels of $[O_3]$ as simulated by the vegetation model. Simulations are performed at 40 North American Carbon Program (NACP) sites with a fixed $[O_3]$ for either low or high $O_3$ sensitivity. The short blue lines show the damages ranging from low to high $O_3$ sensitivity, with the blue points indicating the average reductions. The simulation results are averaged for all the sites or for the sites with the same PFT. The number of sites used for average is shown in the title bracket of each subplot. The solid squares with lines show the results (mean plus uncertainty) based on measurements reported by multiple literatures. For more details, please refer to Yue and Unger (2014).

*l. 43 the authors will capture only partly the feedbacks since ocean temperatures are fixed SST modelling set-up.*

→ Yes, we are limited to fixed SST for the difficulty in the configuration of ocean heat flux, large uncertainty of ocean-atmosphere interaction, and the step-by-step strategy of research. Please refer to our responses to the major comments.

*l. 45 How much are these direct emissions and how does it compare to the feedback effects?*

→ We added values of direct emissions as suggested "Our results suggest that future wildfire may accelerate boreal carbon loss, not only through direct emissions increasing from 68 Tg C yr$^{-1}$ at present day to 130 Tg C yr$^{-1}$ by midcentury, but also through the biophysical impacts of fire aerosols." (Lines 47-50)

*L 55 see l. 45. What is found in this study and how does it relate to the air pollution change in carbon budget?*

→ We have added the number of direct fire emissions in the abstract as suggested (see the above response).

*L68: more uncertain- this is a value judgement in reality we also do not know the ozone impact well either. Perhaps what the authors want to say is that the potential impact is even larger, and can swap sign.*

→ Yes, ozone effect is uncertain in magnitude (species dependent) but is generally negative. We use the statement 'more uncertain' here to indicate that aerosol impact on photosynthesis may change signs at certain conditions.

*L81: on the other hand: to me it looks quite consistent when considering the uncertainties.*

→ The statement has been removed.

*l. 82-95: part of the differences can be due to just using different climate scenarios, and are more or less comparing apples and pears.*

→ We added some results from our previous study to support the conclusion:
"The increasing rate in Balshi et al. (2009) is higher than that in Amiro et al. (2009), indicating substantial uncertainties in fire projections originating from both fire models and simulated future climate. However, even with the same fire models and climate change scenario, large uncertainties (in both magnitude and signs) are found in the projection of area burned among individual climate models (Moritz et al., 2012; Yue et al., 2013)." (Lines 94-99)

*l. 95: perhaps some words why A1B- and how it maps to RCPs (I think it is RCP6.0 equivalent). In the discussion you mention that the various scenarios until 2050 it is statistically almost similar, in my experience it is the 2050s where scenarios start diverging.*

→ The 2050 $CO_2$ concentration is projected to 532 ppm in the A1B scenario, similar to the value of 541 ppm in the RCP8.5 but higher than the value of 478 ppm in the RCP6.0. In method section 2.2, we explain the connection between A1B and RCP8.5 scenarios as follows:

"In the A1B scenario, $CO_2$ concentration is projected to 532 ppm by the year 2050, similar to the value of 541 ppm in IPCC RCP8.5 scenario (van Vuuren et al., 2011) archived for the Coupled Model Intercomparison Project Phase 5 (CMIP5)." (Lines 199-202).

*L 113- this is a very short description. What sensitivity was included? How is consistency between the atmospheric model and land model ensured, how are fluxes calculated? Is Sitch still reflecting the newest knowledge? One can write a whole paper on what is here cryptically mentioned in one sentence.*

→ The Sitch et al. (2007) scheme has been fully evaluated in our previous researches (Figures R1 and R2). We simplify our description here only to emphasize our main focus of this study, which is to examine impacts of fire pollution on ecosystem productivity. In the revised text, we explained more details:

"The impacts of the boreal fire $O_3$ on forest photosynthesis are predicted using the flux-based damage algorithm proposed by Sitch et al. (2007), which has been fully evaluated against available $O_3$ damaging measurements globally and over North America (Yue and Unger, 2014; Yue et al., 2016; Yue et al., 2017)." (Lines 125-128)

"An interactive flux-based $O_3$ damage scheme proposed by Sitch et al. (2007) is applied to quantify the photosynthetic responses to ambient $O_3$ (Yue and Unger, 2014). For this scheme, $O_3$ damaging level is dependent on excess $O_3$ stomatal flux within leaves, which is a function of ambient $O_3$ concentration, boundary layer resistance, and stomatal resistance. Reduction of photosynthesis is calculated on the basis of plant functional types (PFTs), each of which bears a range of low-to-high sensitivities to $O_3$ uptake." (Lines 288-294)

"… simulations F10O3 and F50O3 calculate offline $O_3$ damage based on the simulated $O_3$ from all sources including fire emissions. For these simulations, reductions of GPP are calculated twice with either low or high $O_3$ sensitivity. However, both of these GPP changes are not fed back into the model to influence carbon allocation and tree growth." (Lines 308-311)

"With the Sitch et al. (2007) scheme, the YIBs model simulates reasonable GPP responses to [$O_3$] in North America (Yue and Unger, 2014; Yue et al., 2016). Generally, damage to GPP increases with the enhancement of ambient [$O_3$], but with varied sensitivities for different plant species (see Fig. 6 of Yue and Unger (2014)). In responses to the same level of [$O_3$], predicted $O_3$ damages are higher for deciduous trees than that for needleleaf trees, consistent with observations from meta-analyses (Wittig et al.,

1023
1024

1025

**Figure 6.** Predicted

show the damages

broadleaf forest), (c)

are performed with t

damages back to af

results are averaged

$(\frac{1}{2}(G10ALI$

The average value

subpanel. Significant

1026

1027

1028

1029

1030

1031

1032

1033

**Figure R2.** Evaluation of O$_3$ damaging scheme over China. For more details, please refer to Yue et al. (2017).

1034

1035

*l. 138-142 most relevant to discuss performance of MODIS retrieval over boreal areas even if it doesn't coincide with the flux sites. Summarize here what Strada found?*

→ We summarized the findings by Strada et al. (2015) as follows:

"Strada et al. (2015) used ground-based AOD observations from the Aerosol Robotic Network (AERONET) near AMF sites to validate the sampling technique of MODIS 3-km AOD product. They found high correlations of 0.89-0.98 and regression slopes from 0.89 to 1.03 for daily AOD between AERONET and MODIS at four AMF sites." (Lines 153-157).

*l. 162 would be good to provide the statistics in the supplementary and give summary*

*here. It is really hard to understand here what is meant with 'much fewer' and how it can still be used.*

→ The number of sample pairs has been shown in Table 4. We added these numbers to the revised text:

"At the two selected sites, we calculate the Pearson's correlation coefficients between half-hourly GPP and different components of PAR. In total, we select 2432 and 3201 pairs of GPP and PAR measurements at CA-Gro and CA-Qfo, respectively." (Lines 146-148)

"In total, we select 65 pairs of GPP and AOD at CA-Gro site and another 59 pairs at CA-Qfo site. The GPP-AOD sampling pairs are much fewer than GPP-PAR, because …" (Lines 176-178)

*l. 160-190 for clarity: future burning is assumed to be depending on fire-weather alone (regression relationship). Is there a relationship of fuel load with CO2 and fire management, if not what could be the possible uncertainties from these assumptions? Not clear here if the climate simulations would include a feedback on fires via the fire weather risk.*

→ We do not consider changes in fuel load due to large uncertainties in the projection. However, we include response of fuel moisture to climate change. We clarified as follows: "As in Amiro et al. (2009) and Yue et al. (2015), we apply constant fuel load for both present day and midcentury because opposite and uncertain factors influence future projections (Kurz et al., 2008; Heyder et al., 2011; Friend et al., 2014; Kim et al., 2017). Instead, we consider changes in burning severity due to perturbations in fuel moisture as indicated by CFWI indexes (Yue et al., 2015). On average, we estimate a 9% increase in fuel consumption over boreal North America by the midcentury, because higher temperature and lower precipitation result in a future with drier fuel load (Flannigan et al., 2016)." (Lines 237-244)

We discuss the uncertainties of our consumptions in the section 4.2:
"We apply constant land cover and fuel load for both present day and midcentury, but we estimate an increase in fuel consumption due to changes in fuel moisture. Future projection of boreal fuel load is highly uncertain because of multiple contrasting influences. For example, using a dynamic global vegetation model (DGVM) and an ensemble of climate change projections, Heyder et al. (2011) predicted a large-scale dieback in boreal-temperate forests due to increased heat and drought stress in the coming decades. On the contrary, projections using multiple DGVMs show a widespread increase in boreal vegetation carbon under the global warming scenario with $CO_2$ fertilization of photosynthesis (Friend et al., 2014). In addition, compound factors such as greenhouse gas mitigation (Kim et al., 2017), pine beetle outbreak (Kurz et al., 2008), and fire management (Doerr and Santin, 2016) may exert varied impacts on future vegetation and fuel load. Although we apply constant fuel load, we consider changes of

fuel moisture because warmer climate states tend to dry fuel and increase fuel consumption (Flannigan et al., 2016). With constant fuel load but climate-driven fuel moisture, we calculate a 9% increase in boreal fuel consumption by the midcentury (Yue et al., 2015). Although such increment is higher than the prediction of 2-5% by Amiro et al. (2009) for a doubled-$CO_2$ climate, the consumption-induced uncertainty for fire emission is likely limited because changes in area burned are much more profound." (Lines 624-640)

*l. 194 please give the values. What is meant with much higher?*

→ We clarified as follows:
"We use the average value of 1.6 g NO per Kg dry mass burned (DM) from six studies as $NO_x$ emission factor, because the number of 3.0 g NO per Kg DM reported in Andreae and Merlet (2001) is much higher than that of 1.1 g NO per Kg DM from field observations (Alvarado et al., 2010)." (Lines 248-251)

*l. 228-232 Give a short summary on what the flux scheme is about. Summarize in a few lines what was the outcome of this benchmarking, and the consequence for this study.*

→ We clarified as follows:
"An interactive flux-based $O_3$ damage scheme proposed by Sitch et al. (2007) is applied to quantify the photosynthetic responses to ambient $O_3$ (Yue and Unger, 2014). For this scheme, $O_3$ damaging level is dependent on excess $O_3$ stomatal flux within leaves, which is a function of ambient $O_3$ concentration, boundary layer resistance, and stomatal resistance. Reduction of photosynthesis is calculated on the basis of plant functional types (PFTs), each of which bears a range of low-to-high sensitivities to $O_3$ uptake." (Lines 288-294)

*l. 252 how does the RCP8.5 scenario how link to the use of the A1B scenario mentioned earlier (l 95).*

→ We explained the link between RCP8.5 and A1B scenario as follows:
"In the A1B scenario, $CO_2$ concentration is projected to 532 ppm by the year 2050, similar to the value of 541 ppm in IPCC RCP8.5 scenario (van Vuuren et al., 2011) archived for the Coupled Model Intercomparison Project Phase 5 (CMIP5)." (Lines 199-202)

*l. 255 can a short description of the practical implications coming from the climate scenarios be given.*

→ We added the following descriptions:
"Decadal average monthly-varying SST and sea ice of 2006-2015 are used as boundary

conditions for present-day (2010s) runs while that of 2046-2055 are used for future (2050s) runs. In the RCP8.5 scenario, global average SST increases by 0.62 °C while sea ice area decreases by 13.8% at the midcentury compared to the present-day level." (Lines 322-325)

*l. 258 does CO2 impact fires and fire emissions?*

→ We explained as follows:
"The enhancement of $CO_2$ will affect climate (through longwave absorption) and ecosystem productivity (through $CO_2$ fertilization), but not the fire activity and related emissions directly." (Lines 328-330)

*l. 260 Explain better the model set-up: if area is burnt, does that also change the land-cover? Would that contradict the use of prescribed landcover?*

→ We do not predict changes in land cover as multiple factors interplay and offset.
"As a result, a land cover dataset derived from satellite retrievals (Hansen et al., 2003) is applied as boundary conditions for both the 2010s and 2050s." (Lines 336-338).

"We apply constant land cover and fuel load for both present day and midcentury, but we estimate an increase in fuel consumption due to changes in fuel moisture. Future projection of boreal fuel load is highly uncertain because of multiple contrasting influences. For example, using a dynamic global vegetation model (DGVM) and an ensemble of climate change projections, Heyder et al. (2011) predicted a large-scale dieback in boreal-temperate forests due to increased heat and drought stress in the coming decades. On the contrary, projections using multiple DGVMs show a widespread increase in boreal vegetation carbon under the global warming scenario with $CO_2$ fertilization of photosynthesis (Friend et al., 2014). In addition, compound factors such as greenhouse gas mitigation (Kim et al., 2017), pine beetle outbreak (Kurz et al., 2008), and fire management (Doerr and Santin, 2016) may exert varied impacts on future vegetation and fuel load." (Lines 624-634)

*l. 273-274 2 years spin-up and 10 years seems to be a short time scale for ecosystem responses. Can the authors comment to what extent this represents full response.*

→ As we showed below (Figure R3), NPP in four offline simulations reaches equilibrium within a short period, suggesting that a two-year spin-up is enough for the offline simulations.

[Figure]

**Figure R3.** Simulated annual NPP over boreal North America at (a) 2010s and (b) 2050s.

*l. 235-275 I would like to see a description of how the Yale model is treating regrowth after fires fires, and how the dynamics work out on time scales longer than 10 years. Can we expect an interaction between changes in age-structure and ozone and aerosol effects?*

→ The YIBs model does not simulate vegetation dynamics (changes in PFT distribution), but does simulate changes in LAI, growth and tree heights. Please see Response to Reviewer (1) for a full description of our simplified treatment of fuel availability on fire spread in present and future. To our knowledge, there is no available measurement data on age-structure and ozone and aerosol effects, and as such they are not considered here. These types of "second order" interactions will need to be addressed in future research (5-10 year plan) as the coupled chemistry-carbon-climate models advance.

In the revised paper, we emphasize the current limitations of the YIBs model:

"YIBs is a process-based vegetation model that dynamically simulates changes in leaf area index (LAI) through carbon assimilation, respiration, and allocation for prescribed PFTs." (Lines 278-279)

"The YIBs vegetation model cannot simulates changes in PFT fractions. … As a result, a land cover dataset derived from satellite retrievals (Hansen et al., 2003) is applied as boundary conditions for both the 2010s and 2050s." (Lines 331-338)

*l. 287 can you show in supplementary the interpolated fields for the relevant time periods?*

→ Gridded GPP and AOD from observations have been shown in Figures 3c and 3d with a resolution of 2°×2.5°.

*l. 300 would such a light saturation still be valid under changing CO2 conditions? Please*

*comment, and what could be the impact.*

→ We do not have available field observations under changing $CO_2$ conditions, and as a result, we cannot derive the GPP-PAR$_{dir}$ relationships under the changing $CO_2$ conditions. Increased $CO_2$ enhances GPP but inhibits stomatal conductance. These effects may affect light responses of photosynthesis with unclear extents.

*l. 311 Correlation of AOD and GPP is weak to very weak. The value 3.5+/- 1.1 is just the average of the two slopes? What is the meaning of 1.1 is it one standard deviation based on two observation sets?*

→ Yes, the correlation between AOD and GPP is weak at the site CA-Gro but significant at the site CA-Qfo. The poor data availability limits our exploration of AOD-GPP relationships in boreal region. Here, we calculate the average of slopes at sites CA-Gro and CA-Qfo. The value 1.1 is not standard deviation but the range of slopes between two sites. We clarified in the paper as follows: "On average, GPP sensitivity (denoted as mean ± range) is estimated as $3.5 \pm 1.1$ µmol m$^{-2}$ s$^{-1}$ per unit AOD at lower latitudes of boreal regions in the summer." (Lines 388-389)

*l. 323 Indeed patterns look everywhere reasonable except the western part. What could be the cause of this. Any indication on MODIS data quality? Or missing sources in the NASA model that can explain this? Volcanoes?*

→ We plotted AOD from Multi-angle Imaging SpectroRadiometer (MISR) in Figure R4. Similar to MODIS, the MISR AOD also shows high values in western Canada. "The simulation fails to capture the high values in the west, possibly due to a climate model underestimation of biogenic secondary organic aerosol, which may be an important contribution over the western boreal forest." (Lines 401-403)

[Figure]

**Figure R4.** Observed summer AOD from MISR

*l. 326 What is compared here? 24 hr mean over June, July, August? Did the authors compare at the measurement altitude? I would recommend to focus on daytime values, as more relevant for ozone damage and usually less local conditions .The Sitch approach requires fluxes, and the methodology in this paper needs to be described as well.*

→ We changed the validation from 24-hour mean [O3] to maximum daily 8-hour average (MDA8) [O3] in Figure 4. The MDA8 [O3] is a common metric to represent daytime [O3]. For Sitch's scheme, we have explained how it works in the method section 2.3: "For this scheme, $O_3$ damaging level is dependent on excess $O_3$ stomatal flux within leaves, which is a function of ambient $O_3$ concentration, boundary layer resistance, and stomatal resistance. Reduction of photosynthesis is calculated on the basis of plant functional types (PFTs), each of which bears a range of low-to-high sensitivities to $O_3$ uptake." (Lines 290-294). We also showed ozone stomatal flux in Figure 8.

*l. 331-341. The increase in emissions needs to be described here. How is the contribution of wildfire emissions present determined (zero-out?). It seems that the % increase in ozone scales near-linear with the NOx (and other) emissions? But the contribution to AOD much less due to the abundance of secondary organics from BVOC emissions?*

→ We explained more details about fire emissions as follows:
"During 1980-2009, wildfire is observed to burn $2.76 \times 10^6$ ha and 156.3 Tg DM every year over boreal North America. Similarly, the ensemble prediction with fire regression models estimates present-day area burned of $2.88 \times 10^6$ ha yr$^{-1}$ and biomass burned of 160.2 Tg DM yr$^{-1}$ (Yue et al., 2015). By the midcentury, area burned is projected to increase by 77% (to $5.10 \times 10^6$ ha yr$^{-1}$) in boreal North America, mainly because of the higher temperature in future fire seasons. Consequently, biomass burned increases by 93% (to 308.6 Tg yr$^{-1}$) because fuel consumption also increases by 9% on average in a drier climate (Yue et al., 2015)." (Lines 434-441)

The contribution of wildfire emissions is calculated as: fire-induced air pollution / (background plus fire-induced air pollution) × 100%. As a result, the fire-induced air pollution is not zero out in the denominator.

We showed absolute changes of ozone and aerosols as follows:
"On average, wildfire emissions contribute $7.1 \pm 3.1\%$ ($2.1 \pm 0.9$ ppbv) to surface $O_3$ and $25.7 \pm 2.4\%$ ($0.03 \pm 0.003$) to AOD in the summer over boreal North America in the present day. By midcentury, these ratios increase significantly to $12.8 \pm 2.8\%$ ($4.2 \pm 0.9$ ppbv) for $O_3$ and $36.7 \pm 2.0\%$ ($0.05 \pm 0.003$) for AOD." (Lines 445-448) As it shows, absolute change of AOD is less than $O_3$, which is not relate to the abundance of BSOA.

Changes of $O_3$ and AOD are not only dependent on emissions, but also on chemical processes and physical deposition. In a warmer climate, production of $O_3$ is faster, which may in part explain why $O_3$ enhancement is higher than AOD. In addition, atmospheric

circulation may cause different diffusion for $O_3$ and aerosols due to their different mass load. As a result, we cannot conclude that $[O_3]$ is linear to emissions while aerosol is non-linear.

*l. 344-355 the ozone damage discussion is extremely handwaving and confusing. Where is the 40 ppb threshold coming from, and how does that compare to the use of the Sitch method? Only from Figure 4 I understand that indeed the Sitch high and low sensitivities have been used, but it is not discussed in the text. Anyway it seems that the model has a lot of data point above 40 ppb- but it hard to figure out how good the model performance is where the fire emissions have an impact.*

→ We added new statement and analyses in the revised paper (see the responses to the major comments) to support our conclusions about ozone effects. Here, we explained why 40 ppbv is used as a threshold: "Over boreal North America, dominant PFTs are ENF (accounting for 44% of total vegetation cover) and tundra (treated as shrubland, accounting for 41% of total vegetation cover). Both species have shown relatively high $O_3$ tolerance with a damaging threshold of 40 ppbv as calculated with Sitch's scheme (Yue and Unger, 2014)" (Lines 457-460)

*l. 357 at this point it is not clear what optical properties have been assigned to particles.*

→ We explained in the method section 2.3 about the optical properties for aerosols:
"Size-dependent optical parameters computed from Mie scattering, including extinction coefficient, single scattering albedo, and asymmetry parameters, are applied for each aerosol type (Schmidt et al., 2014)." (Lines 267-270)

*l. 369-381 The circulation feedbacks are an important result of the paper, but due to the approach of constraining SST will include only part of the feedbacks. I would argue that the authors should try to address in one additional simulation why they can ignore these longer timescales.*

→ Please see our response to the major comments.

*l. 402-414 We shouldn't expect a full attribution of feedbacks due to aerosol- so this is pretty convincing. However, as soil moisture is the most important feedback- I am missing here completely a discussion on how realistic soil moisture is represented in the current modeling system, and how the soil moisture feedback is leading to increased burning. At this moment a discussion of the short effects on the carbon cycle by increased burning is missing.*

→ We evaluated the baseline simulation of soil moisture in Fig. S1. For this study, we do not consider the feedback of soil moisture on biomass burning. The fire prediction is

performed independently by considering impacts of temperature, relative humidity, and fire indexes (Yue et al., 2013; Yue et al., 2015).

Reviewer (1) also commented on the many possible interactions among climate, fire, and carbon cycle. However, Reviewer (1) suggests to clarify the main chain of events to reduce the complexity and uncertainty of the analyses: "*Given this enormously complex web of causes and effects, I am not sure what we really learn here. It us up to the authors to clarify and give us a clear picture of what this paper is really about.*" By considering the opinions of both reviewers, we plotted the new figure 1 to illustrate the main processes examined.

*L 433 In Amazonia a large fraction (perhaps more than 50 %) is due to deforestation fires, and may not have a link to soil moisture. Discuss*

→ The comparison here is for the aerosol diffuse fertilization. Both studies compare the changes of carbon fluxes by perturbations of diffuse radiation induced by fire aerosols. No effects of soil moisture are included.

*l. 434 discrepancy or just a difference.*

→ We revised to: "There are at least two reasons for such a difference" (Line 550)

*l. 457 Again- we need to know more about how good soil water is represented in the model- as the paper relies so much on the changes of soil moisture.*

→ We presented the evaluation of soil moisture in Figure S1. The comparisons show that the ModelE2-YIBs model generally reproduces the reasonable spatial pattern with low biases. "For >3300 land grids in the summer, the spatial correlation coefficient is $R = 0.25$ between ModelE2-YIBs and CLM, and $R = 0.34$ between CLM and ERA-Interim. The global area-weighted soil moisture is 0.22 mm$^3$ mm$^{-3}$ for ModelE2-YIBs, 0.26 mm$^3$ mm$^{-3}$ for CLM, and 0.23 mm$^3$ mm$^{-3}$ for ERA-Interim. Statistics for winter are very similar to the summer results."

*l. 493 actual->observed*

→ Changed as suggested.

*l. 524 where is this number coming from.*

→ A similar number of 68 Tg C yr$^{-1}$ is estimated for present day. These numbers are calculated as product of biomass burned and emission factors for $CO_2$ from Andreae and

Merlet (2001). In the paper, we clarified as follows:

"Fire pollution aerosol increases boreal NPP by 72 Tg C yr$^{-1}$ in the present day, comparable to the direct carbon loss of 68 Tg C yr$^{-1}$ from wildfire $CO_2$ emissions (product of biomass burned and $CO_2$ emission factors)." (Lines 673-675)

*l. 402-527 Discussion should better reflect the uncertainties of this work, and contrast them to other climatic effects on the boreal carbon cycle.*

→ We extend the discussion about the uncertainties and caveats of the research: "In this study, we examine the interactions among climate change, fire activity, air pollution, and ecosystem productivity. To reduce the complexity of the interactions, we focus on the most likely dominant feedback and thus main chain of events: "climate → fire → pollution → biosphere' (Fig. 1). However, our choice of feedback analysis does not mean that the interplay of other processes is unimportant. For example, climate-induced changes in vegetation cover/types can influence fire activity by alteration of fuel load, and air pollution by BVOC emissions (climate → biosphere → fire/pollution). In addition, other feedbacks may amplify ecosystem responses but are not considered. For example, the drought caused by fire aerosols in the midcentury (Fig. 11) may help increase fire activity (fire → pollution → climate → fire). Furthermore, we apply fixed SSTs in the climate simulations because reliable ocean heat fluxes for the future world were not available. Many previous studies have investigated regional aerosol-climate feedbacks without ocean responses. For example, Cook et al. (2009) found that dust-climate-vegetation feedback promotes drought in U.S., with a climate model driven by prescribed SSTs. Similarly, Liu (2005) found fire aerosols enhance regional drought using a regional climate model, which even ignores the feedback between local climate and large-scale circulation. While we do concede that our experimental design is not a complete assessment of all known processes and feedbacks, within these limitations, this study for the first time quantifies the indirect impacts of wildfire on long-range ecosystem productivity under climate change." (Lines 581-599)

ModelE2-YIBs represents the full carbon cycle for land ecosystems and terrestrial vegetation, but the current version does not include dynamic atmospheric $CO_2$ or dynamic ocean $CO_2$ cycle. The soil respiration takes thousands of years to reach equilibrium in the model, evolves on much longer timescales than air pollution chemistry (centuries/millennia versus years/decades), and requires transient versus time-slice simulations. Therefore, we made a decision to focus on ecosystem productivity, rather than the longer-term land carbon storage, as our metric of impact. In the discussion, we clarify that NPP is different from NEE but can be used as an indicator for the ecosystem carbon uptake: "
[revised manuscript text omitted]

Yeh, S.-W., Yim, B. Y., Noh, Y., and Dewitte, B.: Changes in mixed layer depth under climate change projections in two CGCMs, Climate Dynamics, 33, 199-213, doi:10.1007/s00382-009-0530-y, 2009.

Yue, X., Mickley, L. J., Logan, J. A., and Kaplan, J. O.: Ensemble projections of wildfire activity and carbonaceous aerosol concentrations over the western United States in the mid-21st century, Atmos. Environ., 77, 767-780, doi:10.1016/J.Atmosenv.2013.06.003, 2013.

Yue, X., and Unger, N.: Ozone vegetation damage effects on gross primary productivity in the United States, Atmospheric Chemistry and Physics, 14, 9137-9153, doi:10.5194/acp-14-9137-2014, 2014.

Yue, X., Mickley, L. J., Logan, J. A., Hudman, R. C., Martin, M. V., and Yantosca, R. M.: Impact of 2050 climate change on North American wildfire: consequences for ozone air quality, Atmospheric Chemistry and Physics, 15, 10033-10055, doi:10.5194/acp-15-10033-2015, 2015.

Yue, X., Keenan, T. F., Munger, W., and Unger, N.: Limited effect of ozone reductions on the 20-year photosynthesis trend at Harvard forest, Global Change Biology, 22, 3750-3759, doi:10.1111/gcb.13300, 2016.

Yue, X., Unger, N., Harper, K., Xia, X., Liao, H., Zhu, T., Xiao, J., Feng, Z., and Li, J.: Ozone and haze pollution weakens net primary productivity in China, Atmospheric Chemistry and Physics, 17, 6073-6089, doi:10.5194/acp-17-6073-2017, 2017.

---

## Author Response (AR2)

*The authors have done a thorough revision of the manuscript, and I would like to thank them for their efforts in accommodating all reviewer comments. In particular the additional model evaluation, improved discussion and change of title have substantially improved the manuscript. I agree with all responses, except for one thing: I do think that impacts of changing fuel load (driven by changing productivity) on fire emissions could be substantial, and could potentially alter the results - even if the uncertainties are currently too large to quantify the effect. For example, Knorr et al. (2016) found a large impact on fire emissions from increasing fuel load due to CO2 fertilization - compensated by varying degrees by decreasing fuel load due to climate change (Fig. 5 there). Since both have substantial uncertainties, we have a situation with two large and uncertain effects compensating each other. The negative impact on fire emissions due to climate change simulated with a DGVM in that paper could be moderated through the aerosol impact on NPP the authors discuss in the present manuscript. This would then constitutes and a negative feedback loop, which could potential dampen the impact of fire emissions on plant productivity. As said, I would like to see an additional discussion of this.*

*Knorr, W., Jiang, L., and Arneth, A.: Climate, CO2, and demographic impacts on global wildfire emissions, Biogeosci., 13, 267-282, doi:10.5194/bg-13-267-2016, 2016.*

RESPONSE: We thank the reviewer for helpful comments. The study of Knorr et al. (2016) mentioned by the reviewer actually supports our conclusion that prediction of future fuel load is very uncertain. On one hand, $CO_2$ fertilization may increase global vegetation carbon (hence fuel load). On the other hand, changes in climate and population may cause opposite trends in fuel load, offsetting $CO_2$ effects. Furthermore, fire prediction in Knorr et al. (2016) is performed based on DGVM, which is not evaluated against observations. Such deficit may introduce uncertainties to the foundation of projection but is hard to quantify. Taken all these together, we consider our assumption of constant fuel load is logically sound, and we have discussed the related uncertainties thoroughly.

In the revised paper, we include Knorr et al. (2016) paper and extend our discussion about uncertainty sources in the projection of fuel load as follows (see underlined words):

[revised manuscript text omitted]

(a) ΔTAS by fire aerosol in 2010

-1.0  -0.6  -0.2  0.2  0.6  1.0 (°C)

(b) ΔTAS by fire aerosol in 2050

-1.0  -0.6  -0.2  0.2  0.6  1.0 (°C)

(c) ΔPREC by fire aerosol in 2010

-1.0  -0.6  -0.2  0.2  0.6  1.0 (mm d⁻¹)

(d) ΔPREC by fire aerosol in 2050

-1.0  -0.6  -0.2  0.2  0.6  1.0 (mm d⁻¹)

(e) ΔSOILW by fire aerosol in 2010

-30  -18  -6  6  18  30 (%)

(f) ΔSOILW by fire aerosol in 2050

-30  -18  -6  6  18  30 (%)

**Figure 11.** Predicted changes in summertime (a, b) surface air temperature, (c, d) precipitation, and (e, f) soil water content at surface caused by aerosols from wildfire emissions at (a, c, e) present day and (b, d, f) midcentury. Results for temperature and precipitation are shown as absolute changes. Results for soil water are shown as relative changes. Results for the 2010s are calculated as (F10AERO - F10CTRL). Results for the 2050s are calculated as (F50AERO - F50CTRL). Significant changes ($p$<0.05) are marked with black dots.

(a) ΔNPP by fire aerosol in 2010    (b) ΔNPP by fire aerosol in 2050

[Figure]

-30    -18    -6    6    18    30    (%)

**Figure 12.** Predicted percentage changes in summer NPP caused by wildfire aerosols at (a)

present day and (b) midcentury. Results for the 2010s are calculated as (F10AERO/F10CTRL

- 1) × 100%. Results for the 2050s are calculated as (F50AERO/F50CTRL - 1)×100%.

Significant changes ($p<0.05$) are marked with black dots.